# Reconstructing cell cycle pseudo time-series via single-cell transcriptome data

Zehua Liu [1], Huazhe Lou[2], Kaikun Xie[2], Hao Wang[2], Ning Chen[2], Oscar M. Aparicio[3], Michael Q. Zhang[1,4], Rui Jiang[1] & Ting Chen[2,3]

Single-cell mRNA sequencing, which permits whole transcriptional profiling of individual cells, has been widely applied to study growth and development of tissues and tumors. Resolving cell cycle for such groups of cells is significant, but may not be adequately achieved by commonly used approaches. Here we develop a traveling salesman problem and hidden Markov model-based computational method named reCAT, to recover cell cycle along time for unsynchronized single-cell transcriptome data. We independently test reCAT for accuracy and reliability using several data sets. We find that cell cycle genes cluster into two major waves of expression, which correspond to the two well-known checkpoints, G1 and G2. Moreover, we leverage reCAT to exhibit methylation variation along the recovered cell cycle. Thus, reCAT shows the potential to elucidate diverse profiles of cell cycle, as well as other cyclic or circadian processes (e.g., in liver), on single-cell resolution.

[1] MOE Key Laboratory of Bioinformatics, Bioinformatics Division and Center for Synthetic & Systems Biology, TNLIST, Department of Automation, Tsinghua University, Beijing 100084, China. [2] MOE Key Laboratory of Bioinformatics, Bioinformatics Division and Center for Synthetic & Systems Biology, TNLIST, Department of Computer Sciences, State Key Lab of Intelligent Technology and Systems, Tsinghua University, Beijing 100084, China. [3] Program in Computational Biology and Bioinformatics, University of Southern California, Los Angeles, CA 90089, USA. [4] Department of Molecular and Cell Biology, Center for Systems Biology, University of Texas at Dallas, 800 West Campbell Road, RL11, Richardson, TX 75080-3021, USA. Correspondence and requests for materials should be addressed to R.J. (email: ruijiang@tsinghua.edu.cn) or to T.C. (email: tingchen@mail.tsinghua.edu.cn)

Cell cycle studies, a long-standing research area in biology, are supported by transcriptome profiling with traditional technologies, such as qPCR[1], microarrays[2], and RNA-seq[3], which have been used to quantitate gene expression during cell cycle. However, these strategies require a large amount of synchronized cells, i.e., microarray and bulk RNA-seq, or they may lack observation of whole transcriptome, i.e., qPCR. Moreover, in the absence of elaborative and efficient cell cycle labeling methods, a high-resolution whole transcriptomic profile along an intact cell cycle remains unavailable.

Recently, single-cell RNA-sequencing (scRNA-seq) has become an efficient and reliable experimental technology for fast and low-cost transcriptome profiling at the single-cell level[4, 5]. The technology is employed to efficiently extract mRNA molecules from single cells and amplify them to certain abundance for sequencing[6]. Single-cell transcriptomes facilitate research to examine temporal, spatial and micro-scale variations of cells. This includes (1) exploring temporal progress of single cells and their relationship with cellular processes, for example, transcriptome profiling at different time phases after activation of dendritic cells[7], (2) characterizing spatial-functional associations at single-cell resolution which is essential to understand tumors and complex tissues, such as space orientation of different brain cells[8], and (3) unraveling micro-scale differences among homogeneous cells, inferring, for example, axonal arborization and action potential amplitude of individual neurons[9].

One of the major challenges of scRNA-seq data analysis involves separating biological variations from high-level technical noise, and dissecting multiple intertwining factors contributing to biological variations. Among all these factors, determining cell cycle stages of single cells is critical and central to other analyses, such as determination of cell types and developmental stages, quantification of cell–cell difference, and stochasticity of gene expression[10]. Related computational methods have been developed to analyze scRNA-seq data sets, including identifying oscillating genes and using them to order single cells for cell cycle (Oscope)[11], classifying single cells to specific cell cycle stages (Cyclone)[12], and scoring single cells in order to reconstruct a cell cycle time-series manually[13]. Besides, several computational models have been proposed to reconstruct the time-series of differentiation process, including principal curved analysis (SCUBA)[14], construction of minimum spanning trees (Monocle[15] and TSCAN[16]), nearest-neighbor graphs (Wanderlust[17] and Wishbone[18]) and diffusion maps (DPT)[19]. In fact, even before scRNA-seq came into popular use, the reconstruction of cell cycle time-series was accomplished using, for example, a fluorescent reporter and DNA content signals (ERA)[20], and images of fixed cells (Cycler)[21]. However, despite these efforts, accurate and robust methods to elucidate time-series of cell cycle transcriptome at single cell resolution are still lacking.

Here we propose a computational method termed reCAT (recover cycle along time) to reconstruct cell cycle time-series using single-cell transcriptome data. reCAT can be used to analyze almost any kind of unsynchronized scRNA-seq data set to obtain a high-resolution cell cycle time-series. In the following, we first show one marker gene is not sufficient to give reliable information about cell cycle stages in scRNA-seq data sets. Next, we give an overview of the design of reCAT, followed by an illustration of applying reCAT to a single cell RNA-seq data set called mESC-SMARTer, and the demonstration of robustness and accuracy of reCAT. At the end, we give detailed analyses of several applications of reCAT. All data sets used in this study are listed in Table 1.

## Results

**High variation of expression measures within cells.** We found that the expression level of one marker gene was insufficient to reveal the cell cycle stage of a single cell as a result of high stochasticity of gene expression and heterogeneity of cell samples. Therefore, we propose to use a group of cell cycle marker genes, combined with proper computational models, to reconstruct pseudo cell cycles from scRNA-seq data with high accuracy.

Using a mouse embryonic stem cells (mESC) scRNA-seq data set developed by Buttener et al. (2015)[22], we showed that the expression of cell cycle marker genes has high stochasticity. The data set, termed mESC-SMARTer, consists of 232 eligible samples labeled according to cell cycle stages by Hoechst staining. We examined several high-confidence cell cycle marker genes, as shown in Fig. 1a. The cell cycle stages in which these genes have maximum mean relative expression levels are consistent with their existing records[29], but the distribution of expression levels between two cell cycle stages showed high overlap (Fig. 1a), indicating that a single marker gene is insufficient to determine the cell cycle stage for a single cell. In addition, we showed that mean gene expression levels, averaging over 20 cell samples, remain highly stochastic (Supplementary Fig. 1).

We further examined the consistency of cell cycle stages of maximum mean expression levels of cell cycle marker genes between different cell populations. We selected six single-cell transcriptome sample groups from different tissues and experimental conditions (Table 1), and performed four pairwise comparisons, showing the results in Fig. 1b. Assuming consistency between maximum mean expression levels of marker genes and their corresponding cell cycle stages, all drops should be located along the diagonal. In fact, however, many counts spread into off-diagonal entries, showing apparent relatively low consistency (Fig. 1b).

**An overview of the reCAT approach.** Given an scRNA-seq data set, reCAT reconstructs cell cycle time-series and predicts cell

---

**Table 1 A list of the single cell transcriptome data sets.**

| No. | Abbreviation | Tissue | Cell cycle stage labeled |
|---|---|---|---|
| 1 | mESC-SMARTer[22] | mESC | YES |
| 2 | mESC-Quartz[23] | mESC | YES |
| 3 | 3Line-qPCR[24] | H9, MB, PC3 | YES |
| 4 | hESC[11] | hESC | YES |
| 5 | mHSC[13] | Long term-HSC (LT-HSC), short term-HSC (ST-HSC), multi-potent progenitor (MPP) | NO |
| 6 | mESC-Cmp[25] | mESC (in 2i, Serum and a2i medium) | NO |
| 7 | hMyo[15] | Differentiating myoblasts (0 h, 24 h, 48 h, 72 h) | NO |
| 8 | mDLM[26] | Distal lung epithelium cells (E14.5, E16.5, E18.5, adult AT2) | NO |
| 9 | hMel[27] | Human metastatic melanoma (Mel78, Mel79 et al.) | NO |
| 10 | mESC-MT[28] | mESC | NO |

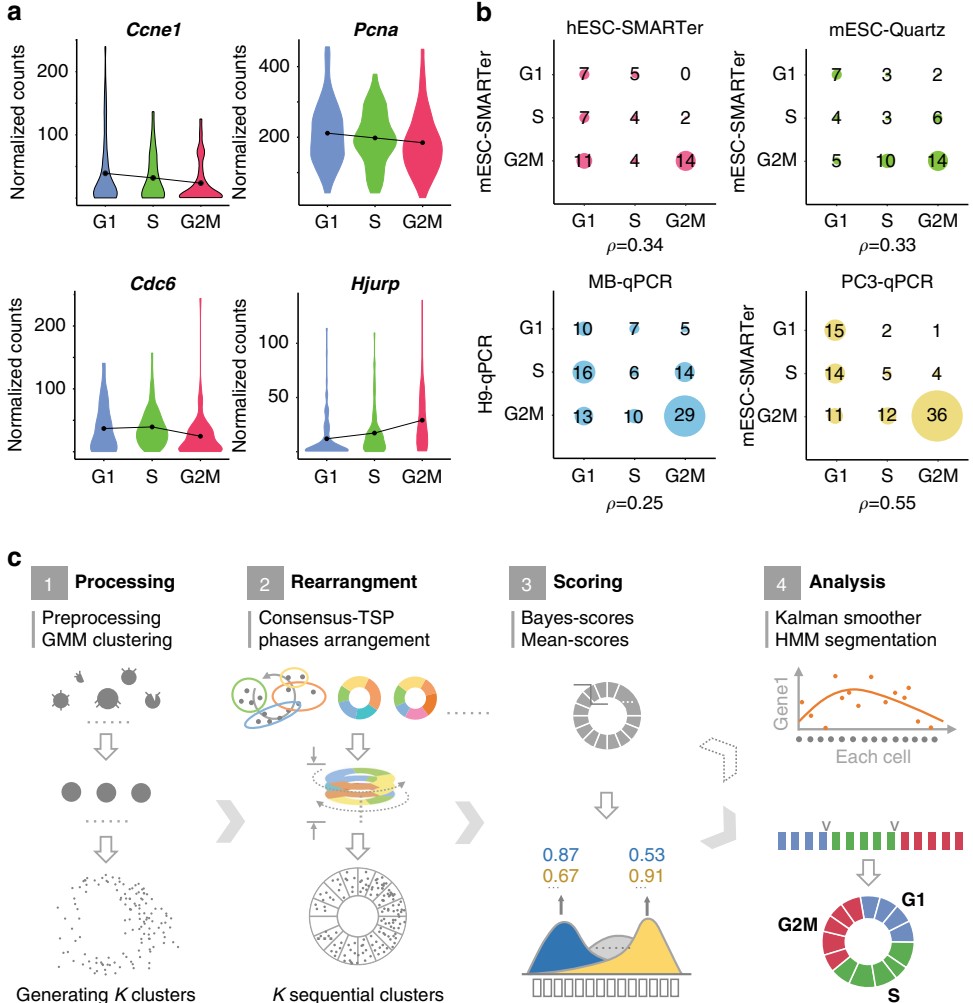

**Fig. 1** High uncertainty of marker gene expression in single cells and the workflow of reCAT. **a** Violin plots of distributions of normalized relative expression levels of cell cycle genes, including *Ccne1*, *Pcna*, *Cdc6* and *Hjurp*, at three stages (G1, S, G2M) using 232 mESCs. **b** A 3 × 3 drop density plot for comparison of two groups of cells based on the number of cell cycle-related genes with maximum mean expression at each of the three cell cycle stages. At the *two top panels*, a set of 60 high-confidence cell cycle related genes (Supplementary Table 1), with a comparable number from each cell cycle stage at which they are known to have maximum expression levels, were used for comparisons. The size of each disk in the matrix is proportional to the number of genes in the entry. The hESC cell group consists of 228 human embryonic stem cells labeled by FUCCI[30], and the mESC-Quartz consists of 21 single mESCs labeled by Hoechst staining for cell cycle stages and sequenced by Quartz-seq[23]. The *two bottom panels* show comparisons among three cell lines, including H9, MB and PC3 from a data set marked as 3Line-qPCR, and mESC-SMARTer. Cells in 3Line-qPCR were labeled by Hoechst staining, and expression levels of 110 cell cycle related genes were measured by qPCR[24]. All 110 cell cycle-related genes were used for the comparisons. Pearson's correlation coefficient (*ρ*) was calculated for each pair of groups. **c** The overview of reCAT

cycle stages along the time-series. The reconstructed time-series generally consists of multiple cell cycle phases (e.g., ≥10), each of which may contain one or multiple cells. Two fundamental assumptions underlie the cell cycle model: (1) different cell cycle phases form a cycle and (2) transcriptome at a certain cell cycle phase would have a smaller difference relative to that of its most adjacent phase compared to a more distant phase. Hence, reCAT models the reconstruction of the time-series as a traveling salesman problem (TSP), which herein finds the shortest possible cycle by passing through each cell/cluster exactly once and returning to the start.

As shown in Fig. 1c, reCAT can be described as a process consisting of four steps. The first step is data processing, including quality control, normalization, and clustering of single cells using the Gaussian mixture model (GMM) according to a user-defined phase number *k*. We defined the distance between two clusters as the Euclidian distance between their means. In the second step, the order of the clusters was recovered by finding a

traveling salesman cycle. Since TSP is a well-known NP-hard problem, we developed a novel and robust heuristic algorithm, termed consensus-TSP, to find the solution. For the third step, we designed two scoring methods, Bayes-scores and mean-scores, to discriminate among cycle stages (G0, G1, S, or G2/M). Finally, in the fourth step, we designed a hidden Markov model (HMM) based on these two scoring methods to segment the time-series into G0, G1, S and G2/M, and a Kalman smoother to estimate the underlying gene expression levels of the single-cell time-series (Methods).

**An illustration of reCAT working principles.** The mESC-SMARTer data set (Buettner, et al. 2015) was used to illustrate the principles underlying the reCAT approach. Only cell cycle genes listed in Cyclebase[31] (378) were used in reCAT to get the expression matrix, while other genes were excluded based on the risk of adding noise to the model. The samples were clustered into

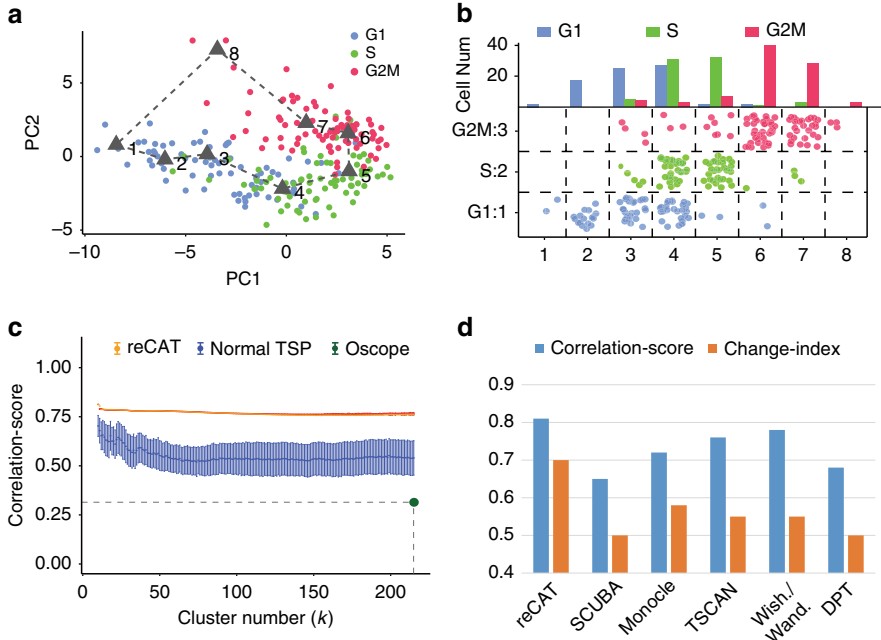

**Fig. 2** Illustration and evaluation of reCAT using the mESC-SMARTer data. **a** PCA visualization of mESC-SMARTer data using expression profiles of the Cyclebase genes (378). Each single cell is colored according to its experimentally determined cell cycle stage. A cycle linking eight *black triangles* represents the shortest traveling salesman cycle of eight cluster means, as computed by reCAT. **b** The bar plot at the *top* shows the composition of single cells at each predicted cell cycle phase. The jitter scatter plot at the *bottom panel* shows how well the ordered clusters are correlated with the known cell cycle stage labels. **c** The correlation-scores for various cluster numbers, between generated cell cycle time-series and experimentally determined cell cycle stage labels, are shown for the consensus-TSP algorithm, the arbitrary insertion algorithm for TSP, and Oscope. For each cluster number '*k*', 200 runs of the arbitrary insertion algorithms were combined to compute and plot means and standard deviations (error bars). **d** Comparison of correlation-scores and change-index values for the time-series generated by reCAT, SCUBA, Monocle, TSCAN, Wish. /Wand., and DPT, using expression of the Cyclebase genes on mESC-SMARTer data

eight classes ($k = 8$), and the mean expression levels of these eight clusters were arranged into the optimal traveling salesman cycle. Fig. 2a displays all single cells and a cycle formed by eight cluster centers in a two-dimensional plot using principal component analysis (PCA), in which *colors* correspond to experimentally determined cell cycle stages. In Fig. 2b, we linearized the traveling salesman cycle into a pseudo time-series of eight phases and plotted the composition of single cells at each phase. The figure shows agreement between the predicted pseudo time-series and the experimentally determined cell cycle stage labels, thereby supporting the validity of the TSP model. In summary, both plots demonstrate a gradual and smooth transition of labeled single-cell components along the pseudo time-series. In the Supplementary Material, we showed that the expression trends of well-studied cell cycle marker genes (Supplementary Table 2) are coherent with the order of the clusters (Supplementary Fig. 2). Moreover, we converted the covariance matrices of each cluster into a vector (Methods) and computed a traveling salesman cycle using these cluster vectors. The generated time-series (Supplementary Fig. 3a) is also consistent with the above one (Fig. 2a), demonstrating that the traveling salesman cycle is inherent within the data.

**Components of reCAT and their validation.** At the center of reCAT is a novel heuristic algorithm, termed consensus-TSP (Methods), to solve TSP robustly. It should be noted that no known polynomial time algorithm can solve the TSP problem for every case. On the other hand, scRNA-seq data are highly noisy; even the optimal traveling salesman cycle may not represent the correct cell cycle order. To overcome these problems, we designed a two-step strategy. In the first step, consensus-TSP groups a set of *n* single cells into *k* clusters for various values of $k \leq n$, and for

each set of *k* clusters, it generates one TSP route using the arbitrary insertion algorithm[32]. Then the second step of consensus-TSP integrates these routes to produce a consensus traveling salesman cycle (Supplementary Fig. 4, Supplementary Note 2).

Consensus-TSP was shown to outperform Oscope[7], the arbitrary insertion algorithm (Fig. 2c), and other well-known TSP algorithms (Supplementary Figs 5 and 6) according to the correlation-score, a Pearson correlation coefficient (PCC)-based scoring function that measures the agreement between a predicted pseudo time-series and experimentally determined cell cycle stage labels (Methods). In Fig. 2d, we demonstrated that consensus-TSP also outperformed current single-cell pseudo time reconstruction methods, including SCUBA[14], Monocle[15], TSCAN[16], Wanderlust[17]/Wishbone[18] and DPT[19] (also in Supplementary Fig. 7 and Supplementary Note 4). The comparisons were based on the correlation-scores and change-index values (Methods). The latter index measures how frequent experimentally determined single cell labels change along the time-series. Consensus-TSP is not only robust (Fig. 2c, Supplementary Note 4) but also scales up well for thousands of single cells (Supplementary Fig. 4f). We observed similar results using the cell cycle stage-labeled mouse embryonic stem cell Quartz-seq (mESC-Quartz) data set[23] (the *left panel* of Fig. 3a) and the cell cycle stage-labeled human embryonic stem cell SMART-seq (hESC) data set[11] (Supplementary Fig. 8a). Of course, the scoring methods of evaluation may have their own limitations. In addition, one point should be noted about the data generation, if cells with the same cell cycle labels were processed and sequenced in the same batch, these cells can be clustered together nicely because of the batch effects, which leads to high scoring values, but cells within each cell cycle stage may not be properly ordered.

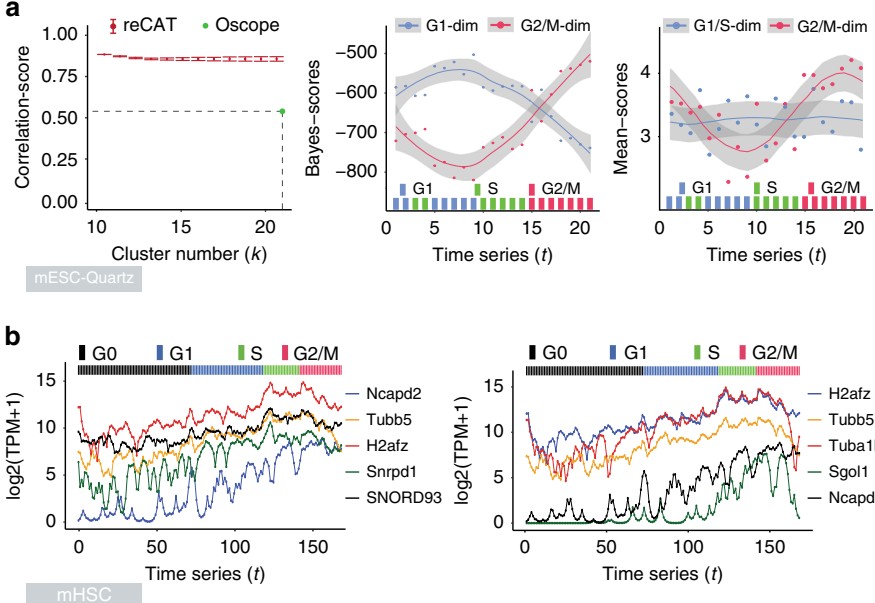

**Fig. 3** Illustration of Bayes-scores, mean-scores and identification of cell cycle-related genes. **a** The *left panel* exhibits a correlation-score curve computed for both reCAT and Oscope (correlation-score: 0.54) using the mESC-Quartz data. The *middle* and *right panels*, respectively display variations of Bayes-scores and mean-scores along the recovered time-series by reCAT. The colored bars at the *bottom* of the *panels* indicate the experimentally determined cell cycle stage labels. **b** Cell cycle associated genes (not in the list of Cyclebase genes) detected from the young-MPP sample cells in the mHSC data. The *left* and *right panel*s show Kalman smoothed TPM (transcripts per million mapped reads) expression levels of top five candidate cell cycle genes, as detected by dCor, and the statistics KNN-MI, respectively. The colored bars at the *top* of the subgraphs indicate cell cycle stages predicted by reCAT

We designed two scoring methods, called 'Bayes-scores' and 'mean-scores', to discriminate among the cell cycle stages (Methods). The Bayes-score is a supervised learning method, which computes Naive Bayesian likelihood values using expression level comparisons of pre-selected gene pairs as input features. The model uses a training data set to determine a fixed number of informative gene pairs[33]. This Naive Bayesian design is able to decrease the effect of stochasticity in scRNA-seq data (Supplementary Fig. 9, Supplementary Note 2). The mean-score is an unsupervised method, which computes the mean of log expression levels of a selected set of marker genes specific to each cell cycle stage. The values of these scores reveal membership of a cluster (or a cell) to a certain cell cycle stage.

We trained the Bayes-scores using the mESC-SMARTer data, and we tested both Bayes-scores and mean-scores on the mESC-Quartz, mESC-SMARTer (only mean-scores) and hESC data sets. The curves of these scores are shown in Fig. 3a, Supplementary Figs 7a and 8b,c, respectively. We observe clear cyclic variations of these curves along cell cycle. In practice, the Bayes-scores performed especially well in distinguishing G0/G1/S from G2/M. The peak for the G1/S mean-score values is usually near the start site of the S stage (Supplementary Figs 7a, 8b and 10), while the peak for the G2/M mean-score values is often near the late G2 stage. For each kind of mean-score, the values at the G0 stage are significantly lower than those at the other stages (Supplementary Note 3), which can be combined into the HMM to discriminate G0 from the other cell cycle stages.

**Identification of cell cycle-related genes.** The noise of gene expression measurements of single cells is high. Therefore, to better observe gene expression variation along the cell cycle time-series computed by reCAT, we designed a Kalman smoother to estimate the sequential expression levels for a gene (Methods). We employed two statistics, distance correlation (dCor)[34] and $K$ nearest neighbors (KNN)-mutual information (KNN-MI)[35], to test the significance of the associations between the

sequential expression levels of a gene and the pseudo time-series, in order to identify cell cycle-related genes not listed in Cyclebase.

We applied the Kalman smoother to the multi-potent progenitor cells from young mice (young-MPP) in the mouse hematopoietic stem cell SMART-seq (mHSC) dataset (Table 1) which contain several groups of mouse hematopoietic stem cells, tested all genes and ranked them according to their significance scores (Supplementary Table 3, Supplementary Fig. 11). Afterwards, the sequential expression levels of the top five non-Cyclebase genes by dCor and KNN-MI were plotted in Fig. 3b. Eight out of the ten genes were confirmed to be strongly related to cell cycle by published literature, although functions of the other two were not clearly recorded (Supplementary Table 4). For instance, *Ncapd2* (non-SMC condensin I complex subunit D2), a protein coding gene, has high expression levels at S and G2 stages (Fig. 3b). It belongs to a large protein complex involved in chromosome condensation, and it is annotated as a cell cycle-related gene by Gene Ontology[36]. However, it was not included in Cyclebase.

**Decomposing proportions of cell cycle stages for mHSCs.** Leveraging Bayes-scores and mean-scores along the pseudo cell cycle time-series, reCAT applies an HMM to segment the time-series into cell cycle stages of G0, G1, S and G2/M (Methods, Supplementary Fig. 12). We applied reCAT to mHSC data, and at the G1 stage, results showed that young individuals had a higher proportion of long-term HSCs (LT-HSC), 41 out of 167 cells, when compared to old individuals with 10 out of 183 cells (Fig. 4a). This is an independent and quantitative confirmation of the original findings by using the staining approach.

**High-resolution transcription atlas of cell cycle in mESCs.** We next applied reCAT to the mESC samples, termed mESC-Cmp,

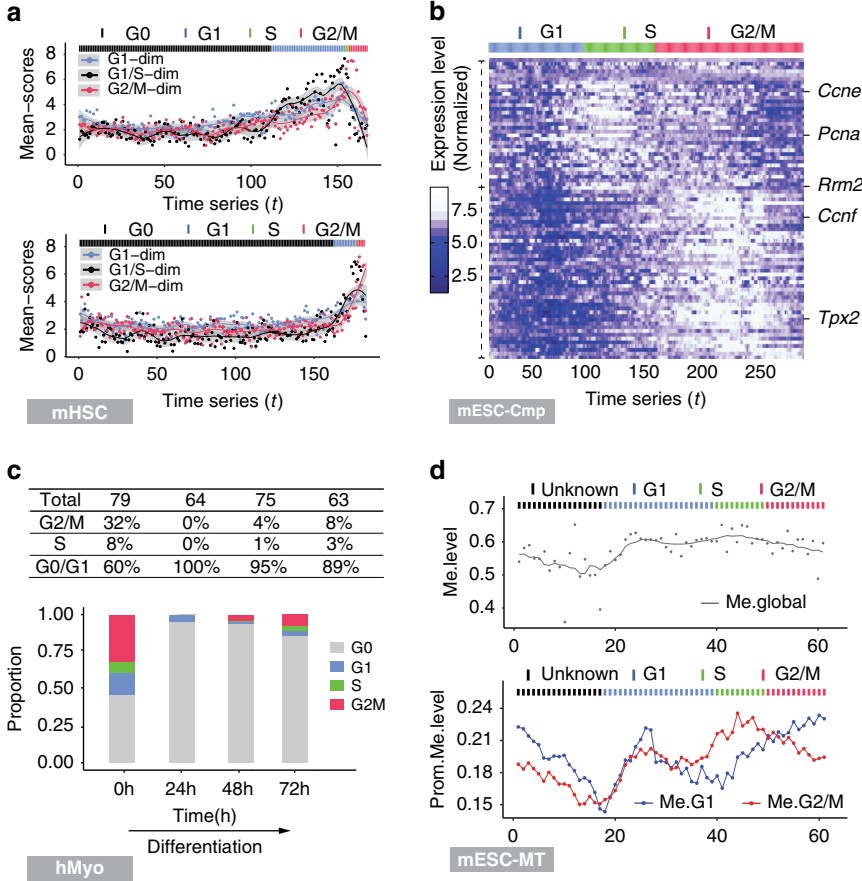

**Fig. 4** Reassessing cell cycle along the recovered pseudo time-series. **a** In the mHSC data, LT-HSCs from young mice have a much higher fraction of G1 stage cells (41 out of 167) than LT-HSCs from old mice (10 out of 183) according to cell cycle stages determined by the HMM model. The curves for G1, G1/S and G2/M dimensions of the mean-scores are plotted along the pseudo time-series. **b** Gene expression along the recovered cell cycle of the mESCs cultured in 2i medium of the mESC-Cmp data. The genes on the vertical axis were arranged according to the recorded peak time from *top* to *bottom*. The data were processed by Kalman smoother, DESeq normalization specific to each gene (not each cell) and logarithm of 2, in order to get better visualization. The *color bars* above each *panel* indicate the cell cycle stages inferred by the HMM model of reCAT. **c** For the differentiating myoblast samples in the hMyo data, four groups of cells were taken at 0th, 24th, 48th, and 72nd hour. reCAT decomposed each sample group into cell cycle stages, and the proportion of each cell cycle stage is shown for each sample group. **d** The *upper panel* shows the smoothed curve for whole genome methylation levels along the pseudo cell cycle time-series of the mESC-MT samples. The *lower panel* shows the smoothed mean methylation levels for promoter regions of Cyclebase-labeled G1 and G2/M peak gene sets

which were cultured in serum, 2i and a2i medium, respectively[25] for comparison (Kolodziejczyk et al. 2015). Previously, Granovskaia et al.[37] built a high-resolution transcription profile using synchronized budding yeast cells. Similarly, we obtained a high-resolution transcription atlas of the mitotic cell cycle in mESCs (Fig. 4b, Supplementary Fig. 13) from scRNA-seq data without synchronization through an in silico approach. Two adjacent cells on the recovered pseudo time-series have a time gap theoretically less than 5 min on average according to the doubling time of about 20 h, which shows a higher resolution than that produced by Granovskaia et al. for budding yeast. During the cell cycles, known cell cycle related genes, arranged by their recorded peak time in Cyclebase (Supplementary Table 5), display two main types of expression waves (Fig. 4b, Supplementary Figs 2 and 13), which correspond to the two well-known checkpoints, G1 and G2. We can also observe decreased expression of cell cycle genes at the end of the cell cycle, which may be caused by degradation of mRNA molecules[38]. We leveraged the decreased expressions to estimate the doubling time of the 2i and serum samples and found it consistent with the values reported in the original paper (Supplementary Fig. 14).

**Changes of stage proportions during differentiation**. We examined scRNA-seq data of human myoblast (hMyo)[15], as developed by Trapnell, et al. (2014), termed hMyo, which consist of differentiating myoblasts sampled at 0th, 24th, 48th, and 72nd hour time points, respectively. We applied reCAT to reconstruct a pseudo cell cycle time-series for each of the four sample groups. Fig. 4c shows the proportions of different cell cycle stages estimated at each sampling time point using the HMM model. A strong negative correlation is shown between differentiation progress and cell cycle activity, as a higher proportion of cells are found in cell cycle at the start of differentiation compared to later differentiation time points. The relatively low proportion of cells in cell cycle at the 72-h time-point is also consistent with the reduced proportion of differentiated cells to divide, as previously documented (Fig. 4c, Supplementary Fig. 15). We obtained a similar result using the mouse distal lung epithelium (mDLM) SMART-seq data set[26], termed mDLM, which consists of four groups of cells sampled at four different developmental stages (Supplementary Fig. 16). In the absence of synchronization procedures during differentiation, each of the four cell groups contains slight inner heterogeneity, further proving that reCAT is unaffected by that factor. Even in a cancer cell data set of human

metastatic melanoma[27], termed hMel, with cancer cell heterogeneity in each sample group, reCAT clearly identified cell cycle status of single cells (Supplementary Fig. 17).

**Recovery of methylation profile along cell cycle**. Using a parallel single-cell genome-wide methylome and transcriptome sequencing data set[28], termed mESC-MT, we show that reCAT is able to recover time-series epigenome along cell cycle via scRNA-seq data. The 61 mESCs were concurrently processed by both SMART-seq for scRNA-seq data and bisulfite sequencing (BS) for single-cell methylation data. We processed the scRNA-seq data first using reCAT to obtain the pseudo time-series (Supplementary Fig. 18) and associated the methylation data with the time-series. We scanned the whole genome methylation levels along the cell cycle (Methods) and discovered that the methylation rate was higher at G1/S phase compared to other cell cycle stages (Fig. 4d). The observation agrees with and extends the conclusion by Brown, et al. (2007)[39], but it contradicts the conclusion of Vandiver, et al. (2015)[40]. Furthermore, we calculated the mean methylation level for promoter regions of gene sets with peak gene expression levels in G1 and G2/M, respectively (Methods). The results imply that the methylation levels for promoter regions of the cell cycle genes vary along cell cycle (Fig. 4d).

## Discussion

Aiming to obtain a high-resolution transcriptomic change that occurs along cell cycle, we developed an scRNA-seq data analysis approach called reCAT. In basic cell cycle studies, reCAT can (1) recover transcriptome change without cell synchronization, which might otherwise alter the native processes, and (2) examine those cells in a developing population or tissue, e.g., during differentiation, that have entered G0 vs. those that continue to divide, thus linking transcriptional changes during development to cell cycle. Therefore, as a novel computational approach to reconstruct cycle along time for unsynchronized single-cell transcriptome data, reCAT is a promising tool with a number of merits. With higher quality and quantity[41] of sequencing samples, more delicate time-series profiles can be modeled in general. Moreover, reCAT has the potential to observe various epigenomics[42, 43] along cell cycle, leveraging parallel sequencing of RNA and DNA[44], which has been demonstrated in this work. Even further, reCAT method can be used in research of other cyclic or circadian expression (e.g., in liver)[45].

reCAT could be refined in several ways. Instead of the pre-selected gene set (378 genes), we would prefer semi-supervised selection of cell cycle genes from the data, as this could lead to better performance in future analysis. The scoring metrics (i.e., Bayes-scores and mean-scores) to indicate cell cycle stages also need improvements to be less noisy and more informative. Additionally, in a given cell cycle, variation of cell cycle-related gene expression predominates over that of the corresponding differentiation. Accordingly, reCAT separates cell cycle analysis from differentiation, which may introduce some bias, but this, too, can be further improved by a combined model. On the contrary, although some reported studies treated cell cycle as noises to be filtered, cell cycle has considerable influence on the investigated biological processes, e.g. myogenesis and embryogenesis. Thus, a model is needed for considering multiple processes simultaneously.

## Methods

**Data set selection**. Ten data sets were used for analysis (Table 1). Among them, four data sets have experimentally derived cell cycle stage labels: the mouse embryonic stem cell RNA-seq data (mESC-SMARTer), mESC-Quartz, hESC, and three cell lines, H9, MB and PC3, sequenced by qPCR. The hESC samples were

labeled by fluorescent ubiquitination-based cell-cycle indicators (FUCCI)[30], while others were labeled by Hoechst staining.

The six unlabeled data sets include mHSC, mESCs scRNA-seq samples from different culture conditions (mESC-Cmp), hMyo cells sampled at four different time points, mDLM cells sampled at four different time points, hMel scRNA-seq samples, and the mESCs processed by scRNA-seq and bisulfite in parallel (mESC-MT). The mHSC, mESC-Cmp and mESC-MT data sets consist homogeneous cells within each group, while the hMyo, mDLM and hMel data sets were sampled from heterogeneous cells.

**Quality control, normalization and preprocessing**. We processed scRNA-seq data in the following procedure. For data with FPKM or TPM expression levels, we considered samples having more than 4000 genes with expression levels exceeding 2, as eligible. For data with counts for expression levels, we followed existing procedures[22] for quality control. Then we deleted genes whose mean expressions were excessively low, e.g., lower than 2 for mean TPM, in order to focus on informative genes. We used the normalization step developed in DESeq[46] to obtain relative expression levels. After quality control and normalization, the expression levels of the 378 cell cycle genes, as defined in Cyclebase, were extracted for downstream analysis. Finally, all gene expression levels were transformed by $\log_2(\text{Exp} + 1)$ to prevent domination of highly expressed genes.

For methylation data, methylation status of a CpG site was considered a binary value in a single cell, unlike a rate in bulk BS. The binary value for single-cell BS data was determined by comparing methylated and unmethylated counts of a CpG site. We generated two results from methylation data of the mESC-MT data set in our analysis. The first result is overall methylation level of whole genome, which is the ratio of the number of methylated sites over the number of all measured sites. The second result is mean methylation levels for promoter regions of two gene sets, which contain Cyclebase genes labeled with G1 and G2/M peak expression, respectively. A gene promoter region was defined as a +/−3 kbp window centered on the transcriptional start site. After methylation levels were obtained, the curves of methylation levels along the pseudo time-series were drawn using an average smoother of nine points.

**Definition of gene sets**. We mainly use four gene sets correlated with cell cycle. (A) The first gene set was obtained from Cyclebase 3.0 which collected 378 genes from dozens of cell cycle-related papers. For genes in Cyclebase, expression peak time, significance and source organisms, for example, are documented. (B) The second set (Supplementary Table 1) consists of 60 highest ranked Cyclebase genes, with 20 having their maximum expression levels at each of three cell cycle stages (G1, S, and G2/M). (C) The third set (Supplementary Table 2) contains 15 high confidence cell cycle related genes selected according to published literatures. (D) The fourth gene set (Supplementary Table 5) includes 120 highest ranked Cyclebase genes, with 20 having their maximum expression levels at each of six cell cycle stages (G1, G1/S, S, G2, G2/M and M).

**Clustering method**. Assume that we are given $n$ single cells, each with an observed expression vector $\mathbf{e}_i = (e_{i1}, \dots, e_{im})$ for $m$ genes and $i = 1,2, \dots, n$. Considering that negative binomial distribution is widely used to model gene expression levels, we approximate the logarithm of the negative binomial distribution by a Gaussian distribution (lognormal). Thus, we used the GMM to model clusters of gene expression profiles of single cells. A GMM with $k$ clusters can be described as:

$$\text{gmm}(\mathbf{e}_i) := \sum_{r=1}^{k} \pi_r \mathcal{N}(\mathbf{e}_i | \boldsymbol{\mu}_r, \boldsymbol{\Phi}_r), \tag{1}$$

where $\mathcal{N}(\cdot | \boldsymbol{\mu}, \boldsymbol{\Phi})$ denotes the Gaussian pdf with mean gene expression vector $\boldsymbol{\mu}$ and covariance matrix $\boldsymbol{\Phi}$, and $\{\pi_1, \dots, \pi_k\}$ are mixture weights satisfying $\sum_{r=1}^{k} \pi_r = 1$ where $0 \leq \pi_r \leq 1$, $r \in \{1, \dots, k\}$. The mixture model can be solved by an expectation maximization algorithm.

**Modeling as a TSP**. We cluster $n$ single cells into $K$ clusters through the GMM whose mean gene expression vectors are $\boldsymbol{\mu}_1, \dots, \boldsymbol{\mu}_K$, each representing a cell cycle phase. Using these $K$ mean vectors, we construct an undirected weighted complete graph $\mathbf{G}$, where nodes correspond to the $K$ mean vectors, and the edges that connect every pair of nodes are weighted by the Euclidean distance between the two vectors. Our goal is to find a Hamilton cycle $\mathbf{C}_K$ in this graph such that every node appears in the cycle exactly once, and the total edge weight of the cycle is minimized. This describes the TSP, which is the classic NP-hard problem in computer algorithm theory.

In our case, the TSP is actually a Euclidean TSP because it satisfies three criteria: non-negative distances, symmetry of distances, and triangle inequality of distances. It should be noted that the Euclidean TSP is also an NP-hard problem, and no known polynomial time algorithm can solve this problem for every case. We therefore designed a heuristic algorithm, called consensus-TSP, which is based on an arbitrary insertion algorithm, to solve the TSP problem[32]. The arbitrary insertion algorithm is a randomized algorithm with $O(n^2)$-running time for a graph with $n$ nodes, and for the worst case, it gives a $2\ln(n)$-approximation. We chose this

algorithm because it can produce a more robust solution than the greedy nearest neighbor algorithm.

Given the generated $K$ clusters, there are two steps for the heuristic TSP algorithm. The first step is to compute traveling salesman cycles for different $k$ (e.g., $k = 7,8, \ldots , K$), and the second step is to merge the cycles into a consensus cycle. In the first step, for each $k$, it takes the $k$ clusters computed from the GMM as input, runs the arbitrary insertion algorithm $n_{fold} \cdot k$ times, and selects the shortest TSP cycle among these $n_{fold} \cdot k$ cycles. In the second step, it merges the $K$ −6 shortest cycles generated in the first step into a consensus-TSP cycle (Supplementary Methods, Supplementary Fig. 4).

**Time-series scoring metrics.** The goal is to develop a quantitative measure of accuracy of computed TSP cycle $\mathbf{C}_k$ using known cell cycle stage labels. Our idea is to compute the PCC between $\mathbf{C}_k$ and experimentally determined cell cycle labels.

Let an $n$-dimensional vector $\bar{\mathbf{l}} = (\bar{l}_1, \ldots , \bar{l}_n)$ denote the experimentally determined cell cycle labels for given $n$ single cells, where $\bar{l} \in \{1, 2, 3\}$ with 1, 2, and 3 indicating the G0/G1, S, and G2/M cell cycle stages, respectively. If cells are labeled by other stages, e.g., G0 or M, the label numbers can be adjusted.

Then we transform the generated traveling salesman cycle $\mathbf{C}_k$ into an $n$-dimensional vector $\mathbf{l}$ as follows. Assume that $\mathbf{C}_k$ consists of a circle of $k$ clusters, $c_1 - c_2 - \cdots - c_k - c_1$. Without loss of generality, we cut the edge $c_k - c_1$ to open the cycle and form a linear path, which represents a pseudo-time series with $c_1$ and $c_k$ as the start and the end of a cell cycle, respectively. We assign a sequential index $j$ to every cell in $j$-th cluster: $l_i = j$ if the $i$-th single cell belongs to the $j$-th cluster along the time-series. Thus we obtain a vector $\mathbf{l} = (l_1, \ldots , l_n)$ where $l_i \in \{1, 2, \ldots , k\}$. We then calculate the PCC between $\bar{\mathbf{l}}$ and $\mathbf{l}$ to measure how well the linear path $c_1 - c_2 - \cdots - c_k$ fits with the experimental data.

Since $\mathbf{C}_k$ has $k$ edges, it can be cut into $k$ different linear paths: $c_1 - c_2 - \ldots - c_k$, $c_2 - c_3 - \ldots - c_k - c_1, \ldots,$ and $c_k - c_1 - \ldots - c_{k-1}$, and their $k$ reverse paths: $c_k - c_{k-1} - \ldots - c_1, c_1 - c_k - c_{k-1} - \ldots - c_2, \ldots,$ and $c_{k-1} - c_{k-2} - \ldots - c_1 - c_k$. For each of these $2k$ paths, we can compute a PCC score and select the maximum PCC score $\rho$ to represent the correlation-score between the traveling salesman cycle $\mathbf{C}_k$ and the experimentally determined cell cycle labels $\bar{\mathbf{l}}$.

The second metric is called "change-index", which measures how frequent an experimentally determined single cell labels changes along the time-series. Ideally, a perfect time-series would change labels twice, G1 to S and S to G2/M. Thus, we define the change-index as $1 - (s_c - 2)/(N - 3)$, where $s_c$ means the sum of the label changes between two adjacent cells. A perfect time-series would have change-index value of 1, while the worst time-series where $s_c = N - 1$ would have a value of 0.

**Bayes-scores and mean-scores to assess cell cycle phases.** Given a traveling salesman cycle $\mathbf{C}_k$ computed from single cell data, we want to determine where the cell cycle stages are located. We designed two methods for this purpose: a supervised Naive Bayes model to compute the probability that a cluster belongs to each of three cell cycle stages, including 'G1', 'S', and 'G2/M' (Bayes-scores), and an unsupervised method to compute the mean expression of a selected subset of cell cycle genes for each of six cell cycle stages, including 'G1', 'G1/S', 'S', 'G2', 'G2/M', 'M' (mean-scores) (Supplementary Methods). Thus Bayes-scores consist of three dimensions and mean-scores consist of six dimensions.

We used the cell cycle-labeled mESC-SMARTer data to train the Bayes-scores. Following the literature[33], we selected a set of informative gene pairs specific to each of the three cell cycle stage; then the gene pairs selected for each stage were unified with $N_p$ pairs (Supplementary Methods). Without loss of generality, we focused on the G1 stage and converted expression of each cluster (or single cell) into a binary vector as follows. For the $i$-th of the $N_p$ pairs, i.e., gene $a$ and gene $b$, we assign a value −1 if their expression levels satisfy $e_a < e_b$, and 1 otherwise. Let the probability $p_i$ be the fraction of G1 stage clusters with value 1 for the $i$-th gene pair, and let the probability $1 - p_i$ be that with value −1. The Naive Bayes model can be expressed as follows: Let $\mathbf{x} = (x_1, \ldots , x_{N_p})$ be the binary vector computed from the gene pairs for an unlabeled cluster. The posterior probability that $\mathbf{x}$ belongs to G1 can be expressed as

$$P(\text{G1}|\mathbf{x}) \propto P(\mathbf{x}|\text{G1})P(\text{G1}) = P(\text{G1})\prod_{i=1}^{N_p} P(x_i|\text{G1}) = P(\text{G1})\prod_{i=1}^{N_p} p_i^{x_i}(1-p_i)^{1-x_i} \tag{2}$$

Thus the Bayes-scores are $\log_{10}(P(\mathbf{x}|\text{G1})P(\text{G1}))$, $\log_{10}(P(\mathbf{x}|\text{S})P(\text{S}))$, and $\log_{10}(P(\mathbf{x}|\text{G2M})P(\text{G2M}))$, respectively, with the prior $P(\text{G1}) = P(\text{S}) = P(\text{G2M})$. We also tested the Lasso-Logistic regression (Supplementary Note 2, Supplementary Methods), but the Naive Bayes had better performance.

To determine the mean-scores of a cluster, which is based on the mean of $\log_2(\text{TPM} + 1)$ of cell cycle genes, we compute the expression mean of a selected subset of marker genes for each cell cycle stage. We selected six gene sets with recorded 'Peaktime' as 'G1', 'G1/S', 'S', 'G2', 'G2/M', and 'M' stage from the Cyclebase genes (378) and then computed the corresponding scores for each cluster (single cell).

**HMM for segmentation.** Given a traveling salesman cycle of $K$ clusters, we applied a HMM (Supplementary Fig. 12) to determine cell cycle stages. Let $\mathbf{H} = \{\text{G0, G1, S, G2/M}\}$ denote the set of hidden states (cell cycle stages) and $\mathbf{A} = (a_{ij})_{N \times N}$ be the

matrix of transition probabilities between the stages, where $N = 4$ denotes the number of stages. If no obvious sign indicates the existence of G0 cells, we only consider G1, S and G2/M. Thus, a state transition exists only when it is from a cell cycle stage to itself or to a physiologically subsequent stage. Along the generated time-series, we characterize a cell $i \in \{1, 2, \ldots , n\}$ using a nine-dimensional scoring vector $\mathbf{o}_i = (o_{i1}, o_{i2}, \ldots , o_{i9})$, which includes three Bayes-scores and the six mean-scores to describe membership of a cell to a specific cell cycle stage. Therefore, when a cell is at a stage $h \in \mathbf{H}$, it emits a nine-dimentional scoring vector described by a multivariate Gaussian distribution $\mathcal{N}(\boldsymbol{\mu}_h, \boldsymbol{\Sigma}_h)$.

Provided with this formulation, we first estimate the parameters $\boldsymbol{\Theta} = (\mathbf{A}, \boldsymbol{\mu}_h, \boldsymbol{\Sigma}_h)$ from the observed scores of cells $\mathbf{O} = (\mathbf{o}_1, \mathbf{o}_2, \ldots , \mathbf{o}_n)$ along the time-series using the Baum–Welch (BW) algorithm. To determine the cell cycle starting point, we tried each cell in the cycle as a starting point, and selected the one that has the highest likelihood for observation. In the implementation of the BW algorithm[47], we adopted logarithm transformation to small intermediate probabilities to avoid underflow. We then implement the Viterbi algorithm to obtain the most likely assignment of the cells, thereby partitioning the time-series into cell cycle stages (Supplementary Methods).

**Kalman smoother and correlation detection.** As scRNA-seq expression noise obeys negative binomial distribution[48], it can be regarded as normal distribution after logarithm. Hence, time-series expression of single cells can be modeled as a random walk plus (RWP) noise model, which is one of the simplest dynamic linear models. Each cell $i$ has a time-series index $t_i \in \{1, 2, \ldots n\}$; hence, the cells can be arranged as $(1, 2, \ldots , T)$ with $n = T$ here. For a selected gene, cells have the observed expression $e_t$ $(t = 1, 2, \ldots , T)$ and the real expression $z_t$ $(t = 1, 2, \ldots , T)$ along the cell cycle time-series. Hence, the RWP model can be expressed as:

$$\begin{cases} e_t = z_t + v, & v \sim \mathcal{N}(0, \sigma_e) \\ z_t = z_{t-1} + w, & w \sim \mathcal{N}(0, \sigma_z) \end{cases} \tag{3}$$

In other words, two adjacent cells have a first-order Markov correlation along the time-series, and the observed expression is generated by adding a normally distributed noise of zero mean to the real expression. In practice, we use Kalman smoother equations, or the Rauch–Tung–Striebel equations (Rauch et al. (1965)) to estimate the real expression $\hat{z}_t$.

With the noise filtered out, we are able to determine whether the expression of a gene exhibits a time-series pattern along the cell cycle by correlating the estimated expression values $\hat{z}_t$ with the time-series index $t$. Apparently, neither Pearson's nor Spearman's correlation coefficients can work here, owing to the non-monotonic property of expression along a time series. Therefore, we adopted three statistical methods (dCor[28], KNN-MI[29], MIC[49]) capable of detecting the nonlinear relationship between two variables.

**Code availability.** The open source implementation of reCAT in R is available on GitHub: https://github.com/tinglab/reCAT.

**Data availability.** No new data was generated in this study. All the data sets used can be find through the accession numbers provided in the original publications cited in Table 1.

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

## Acknowledgements

We thank Xuegong Zhang, Peter Kharchenko, Grace Xiao, Lin Wan and Jianyang Zeng for constructive criticism. We are grateful to Xiangyu Li, Kui Hua, Jun Li, Weilong Guo and Zhiyi Qin for fruitful discussion. We also thank Siqi Qu, Qiongye Dong, Aleksandra A. Kolodziejczyk and Florian Buettner for their technical support. This work was supported by the National Science Foundation of China [61673241, 61561146396], National Basic Research Program of China [2012CB316504, 2012CB316503]; Hi-tech Research and Development Program of China [2012AA020401]; NSFC [61305066, 91010016, 91519326, 31361163004]; NIH/NHGRI [5U01HG006531-03; 4R01HG006465] and the Joint NSFC-ISF Research Program, jointly funded by the National Natural Science Foundation of China and the Israel Science Foundation.

## Author contributions

Z.L. conceived the main strategies and developed the method. Z.L., T.C. and M.Q.Z. designed the study. Z.L., H.L., K.X. and H.W. performed the analysis. Z.L., T.C., R.J., K.X., N.C. and O.M.A. wrote the manuscript.

## Additional information

**Competing interests:** The authors declare no competing financial interests.

