## [Peer Review file · Nature Communications]

Reviewers' comments:

Reviewer #1 (Remarks to the Author):

The authors develop an algorithm to order single cells according to cell cycle through cell cycle gene set transcriptomic profiles. The problem they tackle is of interest in the single cell genomics community as cell cycle classification is an important step in understanding the contributing factors to cellular heterogeneity. The key component is the development of two scoring metrics for the cell cycle stages.

The authors use a Gaussian Mixture Model to perform density estimation and then use a heuristic travelling salesmen problem algorithm to connect the centres to form a cycle. The gene expression measurements are denoised using a Kalman smoother and a Hidden Markov model used to generate the final cell cycle phase assignments. The cell cycle classification algorithm is applied to a number of recently published and well-known datasets.

Overall, I found the organisation of the paper and the quality of the writing challenging. The description of the procedures are not well-structured leading to difficulties in interpretation and potential misconception. In particular, the paper lacks a strong introductory element and enters into the Results section too quickly before establishing the setting of the problem and establishing connection to related literature in the pseudotime estimation literature. There are similarities in methodology to methods such as TSCAN which should be discussed.

Major presentational issues:

Main figures contain analyses from different data sets which needs to be more clearly delineated.

There are 33 Supplementary Figures which seems excessive as some of the analysis appeared repetitive.

There are numerous instances of poor writing such as line 97-98 "reCAT reconstructs cell cycle time-series and predicts cell cycle stages along the time-series".

I was also confused by the derivation of the Bayes scores and Mean-scores. The text states that these are trained using the mESC-SMARTer data. If I have interpreted this correctly this means the results presented in Figure 2 are from an algorithm trained and then tested on the same data? If so, this procedure should be clarified, the algorithm cannot be validated on the same data set it is trained on.

It is also unclear from the plots of the Bayes and mean scores that these truly delineate the different cell cycle stages (e.g. Figure 1D, 3A). They seem quite noisy. I think an important aspect of this work is the evaluation of the robustness of these classification signatures. Some of this analysis is contained in the Supplementary materials but these could be brought into the main text and some of the anecdotal analysis of data sets currently in the main text moved.

The authors use a time-homogeneous Markov model for the cell cycle classification but this assumes the cell states are equally spaced in (pseudo)time, what if the cellular states are non-uniformly spaced in the pseudo temporal cell cycle?

I was confused by the bottom plot of Figure 3B, the authors state in the caption that "The color bars above each panel indicate the cell cycle stages inferred by the HMM model of reCAT" but the colors show illegal transitions between states. Please explain or clarify.

If one of the many recently published pseudotime algorithms is applied to the 378 cell cycle gene set, will they also recover cell cycle ordering? Is there something specific about the way the authors have put together the various steps they have chosen that makes this approach more optimal?

Finally, is the software available for externals? I could not see any links to a public repository. Can the authors demonstrate reproducibility?

Overall, the authors have produced a considerable body of work and aspects of the research are interesting and potentially useful. However, I found the presentation of the work unsatisfactory and there are some key methodological issues highlighted above that need further clarification. I would recommend a substantial revision of the manuscript.

Reviewer #2 (Remarks to the Author):

Summary

In this manuscript, Liu et al develop and describe a computational approach 'recover cell cycle along time' (reCAT) for high-resolution Pseudo-time ordering of asynchronous single cell RNA-Sequencing data (scRNA-seq), utilizing traveling salesman model and further applying Hidden Markov models to segment time-series across distinct cell-cycle stages. The authors use multiple published single cell datasets with known cell-cycle stages to benchmark and test their method and further apply reCAT to asynchronous published single cell data (RNA-seq and methylation) to corroborate previous findings and highlight the power of reCAT to provide high-resolution cell-cycle staging of single cells. The approach would be useful to the community if made accessible and provided the major and minor issues (listed below) are clearly addressed and clarified to strengthen relevant observations and conclusions.

Major issues

- In Figure 1, the authors use a dataset from Buettner et al consisting of 232 single mouse embryonic stem cells (mES) with a priori known cell-cycle stages by Hoechst staining to show that cell cycle marker genes are stochastically expressed. This claim is not correctly addressed as the authors chose to utilize G1 markers to make this point (Fig1a; 3 out of 4 markers are G1 or G1/S) rather than G2M markers that are most highly ranked in CycleBase. The authors seem to highlight cherry picked G1 and S marker genes to highlight stochasticity, poor separation of stages and need for high-

resolution reCAT approach. The validation of reCAT however describes more G2M markers that should inherently perform better than G1 and S markers.

- The author's benchmark reCAT on Hoechst stained mES cell-cycle dataset and further compare with human ES cell-cycle staged dataset for inconsistencies of cell-cycle marker gene expression (Figure 1). The authors should have benchmarked reCAT to more precise cell-cycle staged hES FUCCI dataset and matched with human CycleBase ranking. Subsequent reverse comparison with mESC (Buettner et al and Sasagawa et al) would better highlight the incongruence of cell cycle markers. In addition, the authors in supplementary table 1 should have rather picked top 20 ranked G2M genes and top 20 for G1- and S-phase markers from CycleBase. It is not sure if the chosen 60 genes (Supplementary table 1) are carefully selected or randomly chosen as top periodic genes (especially as the peak times seem to have poor concordance with rank).
- The specific steps involved in reCAT approach are well explained with basic assumptions highlighted. The second assumption for reCAT approach might not necessarily be consistent, especially when transitioning from G1 to S-phase of the cell-cycle. This might also highlight the poor correlation across these stages in single cell dataset.
- In Figure 2a, the authors use travelling salesman to find shortest path (Cyclic) across the 232 mES cells to order the clusters. The Hoechst stained cells (in Buettner et al) should have discreet staging and less continuum that possibly does not capture the cyclic path. Rather the FUCCI hES dataset should be continuous and cyclic progression of cells and a comparison would be very useful.
- In Figure 2e the authors claim that top 5 candidate cell-cycle genes are not present in list of Cyclebase genes. However 4 of the 5 genes are top ranked G2M markers across CycleBase. The selection of genes in the list should be justified.
- Mainstream TSP methods in Figure S10 are incorrectly labeled and ordered and this should be essentially fixed.
- The authors revert back to suggest that Bayes-scores perform well on G2M genes than across G1 and S marker genes. However this is not surprisingly due to the above points (including high ranked G2M markers in Cyclebase and higher expression range). Most mESC dataset used by authors show the same trend of higher correlation of G2M markers and less for other stages. In mESC datasets, there should not be cells in G0 stage.
- In Figure 2e, CDC8A protein is a known cell cycle marker that is reported to have peaktime in G2 stage and well phenotyped across Whitfield dataset. The author's statement should be revised (as its incorrect).
- In Figure 3, the authors compare asynchronous mESC cells and suggest to have made a high-resolution atlas of mitotic cells. This analysis is quite confusing as the authors measure pseudo-time ordering of cells in G2M with doubling time measurement. The doubling time measurement under standard cell culture conditions should be relative similar for asynchronous cells. The percentage of cells across different cell-cycle stages should also be relatively consistent (for cells cultured under same media conditions). In

Figure S27, the figure legend reports that cells are harvested after 2 days but the plot extends for 4 days (96 hours)

- The authors use human myoblast scRNA-seq data from Trapnell et al, who also developed Monocle; Pseudo-ordering of single cells. The authors should compare Monocle and reCAT using same set of cell-cycle genes and highlight if reCAT can better capture the cell-cycle staging and/or resolution as a time series. If reCAT can capture the high-resolution times-series across the different differentiation timepoints (0h, 24h, 48h and 72h) especially with precise cell cycle progression (G0 to G1/S to G2/M), this would be really well received.
- For testing and usability of reCAT, it needs to be made available either as Bioconductor/R- package, standalone source code (Github) or through an online web server. This would make it valuable to the community.

Minor issues

- The text and supplementary figures really needs to be re-ordered. The supplementary figures start from S1 then to S3, S4 further to S12 and so on making it difficult to follow the progression of work.
 - Y-axis on Figure 1b should be reversed (bottom to top; G2M, S and G1;). This would make the expected enlarged nodes off diagonal more intuitive.
 - The choice of 'k' for Gaussian Mixture Model (GMM) is user defined phase number. The authors should comment on how this might be chosen for different datasets (differentiating time-series or relatively heterogenous; Two phase G1/S-G2/M or six phase; G0, G1, G1/S, S, G2,M)
- Figure S23 only has venn plots and not ranked genes according to significance.
 - The authors should retain same sample names/Identifiers across main text, figures and supplementary. There are different instances of cell type, first author name, and protocol all used to represent the same sample.

Point-by-point response to the reviewers' comments

Zehua Liu, KaikunXie, Huazhe Lou, Ting Chen

We thank all the reviewers for their very insightful comments. We have revised the manuscript thoroughly.

Reviewer #1

Q1:

Main figures contain analyses from different data sets which needs to be more clearly delineated. There are 33 Supplementary Figures which seems excessive as some of the analysis appeared repetitive. There are numerous instances of poor writing such as line 97-98 "reCAT reconstructs cell cycle time-series and predicts cell cycle stages along the time-series".

A1:

We carefully addressed these questions by (1) adding a table to list all datasets used in the study, (2) reducing the supplementary figures to 19, and (3) revising the manuscript thoroughly to improve the presentation.

Q2:

I was also confused by the derivation of the Bayes scores and Mean-scores. The text states that these are trained using the mESC-SMARTer data. If I have interpreted this correctly this means the results presented in Figure 2 are from an algorithm trained and then tested on the same data? If so, this procedure should be clarified, the algorithm cannot be validated on the same data set it is trained on.

A2:

As presented in the **Online Methods**, Bayes-scores is a supervised method to train parameters using the mESC-SMARTer data, and mean-scores is an unsupervised method which requires no training data. In **Figure 2a-c**, we illustrated, evaluated the reCAT model, and then applied these two scoring methods to a new dataset called mESC-Quartz in **Figure 2d** and **Figure 2e**. In the revised manuscript, we split these two parts of **Figure 2** into two separate figures.

Q3:

It is also unclear from the plots of the Bayes and mean scores that these truly delineate the different cell cycle stages (e.g. Figure 1D, 3A). They seem quite noisy. I think an important aspect of this work is the evaluation of the robustness of these classification signatures. Some of this analysis is contained in the Supplementary materials but these could be brought into the main text and some of the anecdotal analysis of data sets currently in the main text moved.

A3:

As discussed at the beginning of the paper, the greatest challenge faced in delineating cell cycle stages is the high level of uncertainty of marker gene expression. There, Bayes-scores and mean-scores were designed to overcome this uncertainty. Specifically, Bayes-scores is based on the Naïve Bayesian model that combines expression comparisons of thousands of gene pairs, and mean-scores is based on the means of expression levels of tens of marker genes at each cell cycle stage. In addition, mean-scores has been used as a reliable index in several published articles¹.

Since S stage is a transitive stage, it can easily be confused with G1 and G2/M stages. Therefore, we evaluated our scoring methods by robustness of discrimination between G1 and G2/M cells. We adopted ten-fold cross-validation to test the Bayes-scores and found that 94.3% of G1 cells had higher G1 scores than G2/M scores and that 92.9% of the G2/M cells had higher G2/M scores than G1 scores. For mean-scores, we compared G1/S scores with G2/M scores, and plotted these two scores on a 2-dimensional space in which a linear classifier (generated by SVM) achieved 88.9% accuracy (**Re. I - Figure 1a**). To assess the noise of mean-scores, we

95 compared it with the mean of expression $\log_2(\text{TPM}+1)$ of a set of 29 house-keeping genes^{2,3,4} (MHK) (**Re. I -**
 96 **Table 1**). It is well known that expressions of housekeeping genes are quite stable, hence we leveraged the
 97 property to make a reference to mean-scores. We found that the variance from the regressed cubical curve of
 98 mean-scores (29 genes randomly selected for each gene set) for either G1/S (0.094) or G2/M (0.102) is
 99 comparable to the variance from the regressed quadratic curve of MHK (0.072), as shown in **Re. I - Figure 1b**.

100 **Re. I - Table 1** The selected 29 housekeeping genes.

Gene Symbols					
Chmp2a	Emc7	Psmb2	Psmb4	Reep5	Snrpd3
Vcp	Vps29	Actb	Aip	Cxxc1	Gapdh
Gusb	Hmbs	Hprt	Ipo8	Mrpl48	Mtcp1
Pgk1	Ppia	Rpl13a	Rplp2	Rps6	Tbp
Ubc	Ywhaz	B2m	Hbs1l	Sdha	

103 **Re. I - Figure 1** Assessment for the robustness of mean-scores using the mESC-SMARTer data. (a) Linear
 104 classification of G1 and G2/M cells based on the two mean-scores. (with correct cells of 88.9%) (b) Mean gene
 105 expression levels of three gene groups, as measured by $\log_2(\text{normalized counts} + 1)$, including 29 housekeeping genes,
 106 randomly selected G1/S marker genes, and random selected G2/M marker genes.

107 **Q4:**

108 The authors use a time-homogeneous Markov model for the cell cycle classification but this assumes the cell
 109 states are equally spaced in (pseudo) time, what if the cellular states are non-uniformly spaced in the pseudo
 110 temporal cell cycle?

111 **A4:**

112 We initialized the HMM model with equally spaced (pseudo) cellular states. However, given the iterative EM
 113 (expectation-maximization) computations, the proportions of the cellular states converged to final values that
 114 could be very different from the initial values.

115 **Q5:**

116 I was confused by the bottom plot of Figure 3B, the authors state in the caption that "The color bars above each
 117 panel indicate the cell cycle stages inferred by the HMM model of reCAT" but the colors show illegal transitions
 118 between states. Please explain or clarify.

119 **A5:**

120 It was a mistake and we corrected it in the revised manuscript. The new figures (**Figure 3b**) do not have illegal
 121 transitions. In addition, we show the color bars (cell cycle stages inferred by reCAT) at the top of the panel.

122 **Q6:**

123 If one of the many recently published pseudo time algorithms are applied to the 378 cell cycle gene set, will they
 124 also recover cell cycle ordering? Is there something specific about the way the authors have put together the
 125 various steps they have chosen that makes this approach more optimal?

129 **A6:**

130 We compared reCAT with two recently published methods: TSCAN⁵ and DPT⁶. Similar to Monocle, TSCAN
131 computes a minimum spanning tree (MST) to unravel a linear structure of single cells, and DPT is a diffusion
132 map-based method. In comparison, reCAT computes a traveling salesman cycle that matches a complete cell
133 cycle. We applied these algorithms to two scRNA-seq datasets, mESC-SMARTer and mESC-Cmp, and assessed
134 the results using Bayes-scores and mean-scores, which were developed independently from these pseudo time-
135 series construction algorithms.

136 Using the cell cycle-labeled mESC-SMARTer dataset, we plotted Bayes-scores of the time-series produced
137 by reCAT (**Re. I - Figure 2a**), TSCAN (automatically selected three clusters) (**Re. I - Figure 2c**) and DPT (**Re.
138 I - Figure 2e**), with labeled cell cycle stages at the bottom of each panel. We observed that the G2/M(red) scores
139 of TSCAN and DPT did not decrease at the end of the cell cycle, which is contrary to the biological properties
140 of the G2/M marker genes. To explain, TSCAN and DPT consider linear property of gene expression profiles,
141 not the full cell cycle. We then plotted Bayes-scores of the time-series produced by reCAT (**Re. I - Figure 2b**),
142 TSCAN (automatically selected four clusters) (**Re. I - Figure 2d**) and DPT (**Re. I - Figure 2f**), using the
143 unlabeled 2i samples in the mESC-Cmp data. reCAT produced very clean curves, making it possible to clearly
144 distinguish each cell cycle stage, while the Bayes-scores of TSCAN and DPT time-series are noisy or
145 discontinuous. These results show the superiority of the cell cycle pseudo time-series produced by reCAT.

146 We quantitatively evaluated the pseudo time-series using our 'correlation-score' and 'POS'⁵, in which reCAT
147 outperformed all other methods (**Re. I - Table 2**). However, it should be noted that these two metrics could not
148 accurately measure the order of the time-series. First, different time points generally correspond to distinct
149 experimental batches, and batch effects tend to correlate well with time points because cells from different
150 batches can be easily separated. Second, since the cell cycle labels only distinguish G1, S, and G2/M, these
151 correlation-based scores could not measure accuracy of the cell order within a cell cycle label or a time point.

152 In summary, the algorithm of reCAT is more reliable for cell cycle pseudo-time reconstruction, mainly because
153 of its accuracy and robustness. The good results of reCAT can first be attributed to the fact that it is based on a
154 circular model that brings in more prior information. Second, it merges many routes together to produce a robust
155 result, similar to the Wanderlust approach⁷. Third, it is a nonlinear method able to fit nonlinear properties of data.

156 The hypothesis underlying the reCAT algorithm supposes that the cell cycle can be modeled as a traveling
157 salesman cycle of single cells, in which transcriptome at a certain cell cycle phase would have a smaller
158 difference relative to that of its most adjacent phase compared to a phase more distant from it. Then, GMM-
159 based clustering can reduce stochasticity and noise of single-cell gene expression and produce time-series at
160 varied distinct resolution. Therefore, we combined multiple TSP cycles computed from various GMM clustering
161 resolutions to generate an ensemble time-series that is more robust than any other solutions. Bayes-scores is a
162 key component of reCAT because it leverages the checkpoints during cell cycle and decreases the noise of
163 scRNA-seq dramatically. Mean-scores is also a key component of reCAT because it makes good use of prior
164 knowledge from published literature and decreases the noise of scRNA-seq dramatically. HMM was chosen
165 because it is a segmentation method with fine performance, and the Kalman smoother algorithm can filter
166 stochastic noise along time-series.

167
168
169
170
171
172
173
174
175
176
177
178
179
180

181 **Re. I - Figure 2** Comparison of reCAT, TSCAN and DPT. Mean-scores of the labeled mESC-SMARTer data arranged
 182 by the time-series generated from reCAT (a), TSCAN (c) and DPT (e). Bayes-scores of unlabeled 2i samples in the
 183 mESC-Cmp data arranged by the time-series generated from reCAT (d), TSCAN (d) and DPT (f).

184
 185 **Re. I - Table 2** The 'correlation-score' and 'POS' values of the pseudo time-series by reCAT, TSCAN, Monocle and
 186 DPT, using the mESC-SMARTer dataset.

	Correlation-score	POS
reCAT	0.80	0.84
TSCAN	0.76	0.81
Monocle	0.72	0.77
DPT	0.68	0.74

187
 188 **Q7:**
 189 Finally, is the software available for externals? I could not see any links to a public repository. Can the authors
 190 demonstrate reproducibility?
 191 **A7:**
 192 reCAT is available at <https://github.com/tinglab/reCAT>.

193 **Reviewer #2**

194
195
196
197
198
199
200
201
202
203
204
205
206
207
208
209
210

Q1:

In Figure 1, the authors use a dataset from Buettner et al consisting of 232 single mouse embryonic stem cells (mES) with a priori known cell-cycle stages by Hoechst staining to show that cell cycle marker genes are stochastically expressed. This claim is not correctly addressed as the authors chose to utilize G1 markers to make this point (Fig1a; 3 out of 4 markers are G1 or G1/S) rather than G2M markers that are most highly ranked in CycleBase. The authors seem to highlight cherry picked G1 and S marker genes to highlight stochasticity, poor separation of stages and need for high-resolution reCAT approach. The validation of reCAT however describes more G2M markers that should inherently perform better than G1 and S markers.

A1:

Figure 1a was plotted to illustrate that the stochasticity of expression of cell cycle marker genes is too high to simply make cell cycle stage judgments. G1 and G1/S marker genes were shown in previous experiments to be very important markers of cell cycle stages. Therefore, the original purpose for using more G1 or G1/S genes was based on highlighting the weakness of using a single marker gene in cell cycle stage judgment. The figure below (**Re. I - Figure 3**) includes 8 more G2, G2/M and M marker genes, which display expression stochasticity similar to that of G1/S marker genes.

211 **Re. I - Figure 3** Violin plots of distributions of normalized relative expression levels of G2, G2/M and M marker
212 genes, including *Cdk1*, *Aurka*, *Ccna2*, *Ccnf*, *Kif23*, *Plk1*, *Top2a* and *Tpx2*.

213
214

Q2:

The author's benchmark reCAT on Hoechst stained mES cell-cycle dataset and further compare with human ES cell-cycle staged dataset for inconsistencies of cell-cycle marker gene expression (Figure 1). The authors should have benchmarked reCAT to more precise cell-cycle staged hES FUCCI dataset and matched with human CycleBase ranking. Subsequent reverse comparison with mESC (Buettner et al and Sasagawa et al) would better highlight the incongruence of cell cycle markers. In addition, the authors in supplementary table 1 should have rather picked top 20 ranked G2M genes and top 20 for G1- and S-phase markers from CycleBase. It is not sure if the chosen 60 genes (Supplementary table 1) are carefully selected or randomly chosen as top periodic genes (especially as the peak times seem to have poor concordance with rank).

A2:

We did not use reCAT to process mESC-SMARTer data in **Figure 1**. We benchmarked reCAT on hESC data (cell-cycle staged by FUCCI) and showed the result in **Re. I - Figure 4** (or **Figure S10**). Furthermore, FUCCI may not be able to reach enough accuracy if we want to get more subtle stage labeling⁸ and it is not as accurate as Hoechst staining for only G1, S and G2/M labeling. Genes shown in **Table S1** and **Table S5** are top-ranked genes, but we originally displayed them according to their peak time. In the revised manuscript, we have now rearranged the genes by 'Peakstage' and 'Rank'.

229

231 **Re. I - Figure 4** reCAT recovered cell cycle of the hESC dataset. (a) Correlation-score curves from reCAT and Oscope
 232 (correlation-score: 0.41) results. The Bayes-scores (b) and mean-scores (c) are plotted with respect to the reCAT-
 233 generated time-series at single-cell resolution. The colored bars at the top of the panels indicate the experimentally
 234 determined cell cycle stage labels.

235

236 **Q3:**

237 The specific steps involved in reCAT approach are well explained with basic assumptions highlighted. The
 238 second assumption for reCAT approach might not necessarily be consistent, especially when transitioning from
 239 G1 to S-phase of the cell-cycle. This might also highlight the poor correlation across these stages in single cell
 240 dataset.

241 **A3:**

242 The second assumption for reCAT holds that “the change of gene expression profile from one phase to the next
 243 should be monotonic, increasing with timespan widths”. Here we assume that the whole cell cycle is partitioned
 244 into multiple phases (>4) and that the transcriptome (or expression of a set of genes, rather than an individual
 245 gene) at a certain cell cycle phase would have a smaller difference relative to that of its most adjacent phase,
 246 compared to a phase more distant from it. More simply, this change represents the distance between two
 247 expression profiles, and this assumption establishes the foundation of our traveling salesman problem (TSP)
 248 formulation.

249

250 **Q4:**

251 In Figure 2a, the authors use travelling salesman to find shortest path (Cyclic) across the 232 mES cells to order
 252 the clusters. The Hoechst stained cells (in Buettner et al) should have discreet staging and less continuum, that
 253 possibly does not capture the cyclic path. Rather the FUCCI hES dataset should be continuous and cyclic
 254 progression of cells and a comparison would be very useful.

255 **A4:**

256 According to the experimental procedures of the mESC-SMARTer data, cells were selected continuously⁹.
 257 However, no such description can be found for the hESC data (labeled by FUCCI)¹⁰. Thus, we plot mean-scores
 258 along the recovered pseudo time-series for both datasets in **Re. I - Figure 5a, b** (or **Figure S14b, Figure S13c**).
 259 Apparently, the curves of the mESC data (FUCCI) were more segmented than those of the mESC-SMARTer
 260 data (Hoechst), which may indicate discontinuous selection in the FUCCI experiment.

261

262

263

264

265

266

267

268 **Re. 1 - Figure 5** The mean-scores curves of the mESC-SMRATER (Hoechst) and hESC data (FUCCI). The top color
 269 bars indicate the experimentally generated stage labels. (a) Curves of the mESC-SMRATER (Hoechst). (b) Curves of
 270 the hESC data (FUCCI).

271

272 **Q5:**

273 In Figure 2e the authors claim that top 5 candidate cell-cycle genes are not present in list of Cyclebase genes.
 274 However, 4 of the 5 genes are top ranked G2M markers across CycleBase. The selection of genes in the list
 275 should be justified.

276 **A5:**

277 Thank you. We have already revised the figure.

278

279 **Q6:**

280 Mainstream TSP methods in Figure S10 are incorrectly labeled and ordered and this should be essentially fixed.

281 **A6:**

282 Thank you. We have already revised the legend.

283

284 **Q7:**

285 The authors revert back to suggest that Bayes-scores perform well on G2M genes than across G1 and S marker
 286 genes. However this is not surprisingly due to the above points (including high ranked G2M markers in
 287 Cyclebase and higher expression range). Most mESC dataset used by authors show the same trend of higher
 288 correlation of G2M markers and less for other stages. In mESC datasets, there should not be cells in G0 stage.

289 **A7:**

290 G2/M marker predominance has not been clearly presented in previous studies; therefore, we chose to emphasize
 291 the phenomenon. However, we observed that Bayes-scores performed almost equally well on both G2/M and
 292 G1, G1/S marker genes. We did not find any embryonic stem cells (ESCs) at the G0 stage according to our
 293 standard for stage determination (**Supplementary Note 3**). Nevertheless, we did find some cells in an unknown
 294 status that seemed to be arranged out of the normal cell cycle stages, as shown in **Figure 4d**. Although they were
 295 labeled in black, they are not G0 cells because the means of Cyclebase gene expression, e.g. $\log_2(\text{TPM}+1)$, is
 296 larger than 4 ($m_{c_y} > 4$). These cells may be contaminated or a quiescent state or committed to differentiation.

297

298 **Q8:**

299 In Figure 2e, CDC8A protein is a known cell cycle marker that is reported to have peaktime in G2 stage and well
 300 phenotyped across Whitfield dataset. The author's statement should be revised (as its incorrectness).

301 **A8:**

302 Thank you for pointing this out. We have now changed Cdca8 to Ncapd2.

303

304 **Q9:**

305 In Figure 3, the authors compare asynchronous mESC cells and suggest to have made a high-resolution atlas of
 306 mitotic cells. This analysis is quite confusing as the authors measure pseudo-time ordering of cells in G2M with
 307 doubling time measurement. The doubling time measurement under standard cell culture conditions should be

308 relative similar for asynchronous cells. The percentage of cells across different cell-cycle stages should also be
309 relatively consistent (for cells cultured under same media conditions). In Figure S27, the figure legend reports
310 that cells are harvested after 2 days but the plot extends for 4 days (96 hours).

311 **A9:**

312 The original paper of the data¹¹ aimed to show that cellular transcriptomes of the mESCs, which grew in the
313 three distinct culture conditions, were different. We confirmed with the authors that different culture media could
314 change proportion of cell cycle stages.

315 The authors informed us that they harvested the cells a little bit after 48 hours, although they did not record
316 the exact harvest time. The appearance of “96 hours” is because that they continuously monitored the cell
317 numbers after the harvest time.

318 After communicating with the authors, we also know the cells cultured in serum can be separate into three
319 subpopulations: differentiated-committed mESCs, intermediate mESCs (but not G0), and cell cycle mESCs.
320 However, the cell cycle population is different from the mESCs in the ground pluripotent state, as the authors
321 wrote to us that the expressions of the cell cycle genes were “more homogeneous”, which we also observed, but
322 the reason was not clearly explained. Therefore, we remove the serum figure, in order to exhibit a clear result.
323

324 **Q10:**

325 The authors use human myoblast scRNA-seq data from Trapnell et al, who also developed Monocle; Pseudo-
326 ordering of single cells. The authors should compare Monocle and reCAT using same set of cell-cycle genes and
327 highlight if reCAT can better capture the cell-cycle staging and/or resolution as a time series. If reCAT can
328 capture the high-resolution times-series across the different differentiation time points (0h, 24h, 48h and 72h)
329 especially with precise cell cycle progression (G0 to G1/S to G2/M), this would be really well received.

330 **A10:**

331 In our assessment of Monocle (**Re. I - Table 2**), we showed that reCAT performed better than Monocle. Since
332 the lack of cell cycle stage labels within the hMyo data, we could not directly assess the performance of Monocle,
333 though we did test the hMyo data and found some unevenness in the Bayes-scores and the mean-scores of the
334 time-series generated by Monocle. Therefore, we applied both reCAT and Monocle to the labeled mESC-
335 SMARTer data, and compared the time-series with the cell cycle labels, shown in the color bars at the bottom of
336 the panels of **Re. I - Figure 6a, b**. The accuracy of the time-series computed by reCAT is higher than that by
337 Monocle. Besides, the TSCAN paper⁵ reported that methods based on cell clustering, e.g. TSCAN, or based on
338 ensemble of time-series, e.g. Wanderlust⁷, were generally more robust than Monocle, which indicates the
339 vulnerability of Monocle to disturbance.

340 **Re. I - Figure 6** The pseudo time-series recovered by reCAT (a) and Monocle (b), respectively. The bottom color bars
341 indicate the experimental cell cycle stage labels and the curves are various dimensions of the arranged mean-scores.
342 The color bars of the Monocle result are not well arranged compared to the color bars from reCAT.

343

344 **Q11:**

345 For testing and usability of reCAT, it needs to be made available either as Bioconductor/R-package, standalone
346 source code (Github) or through an online web server. This would make it valuable to the community.

347 **A11:**

348 reCAT is available at <https://github.com/tinglab/reCAT>.

349
350
351
352
353
354
355
356
357
358
359
360
361
362
363
364
365
366
367
368
369
370
371
372
373
374
375
376
377
378
379
380
381
382
383
384
385
386
387
388
389
390
391
392
393
394
395
396
397
398
399
400

Minor issues

Q12:

The text and supplementary figures really needs to be re-ordered. The supplementary figures start from S1 then to S3, S4 further to S12 and so on making it difficult to follow the progression of work.

A12:

We have reordered the supplementary figures. Now each supplementary figure is cited in the main text.

Q13:

Y-axis on Figure 1b should be reversed (bottom to top; G2M, S and G1;). This would make the expected enlarged nodes off diagonal more intuitive.

A13:

This has been fixed.

Q14:

The choice of 'k' for Gaussian Mixture Model (GMM) is user defined phase number. The authors should comment on how this might be chosen for different datasets (differentiating time-series or relatively heterogonous; Two phase G1/S-G2/M or six phase; G0, G1, G1/S, S, G2, M)

A14:

As shown in **Figure 2d**, the correlation score was slightly higher when K is small, and was stabilized for $K > 50$. We tested $K = 10$, $K = 20$ and $K = 50$ on the labeled mESC-SMARTer data (**Re. 1 - Figure 7**). Overall, the mean-scores match their designed properties during cell cycle, and the bottom color bars show that the time series match the cell cycle labels well. However, since larger K gives higher resolution, we usually set ' $K=N$ ' where ' N ' is the number of the cells. For differentiating or relatively heterogonous cells, as shown in the main text, we did not observe apparent difference in the results compared to the homogeneous cell groups. So we will treat them in the same way as those homogeneous cell groups.

Our study shows that transcriptional profiles display three main transitions during M-G1, G0-G1 (if it exists) and S-G2. However, we conformed to traditional segmentation in which the cell cycle is divided into G0, G1, S, G2 and M stages. Since M is typically very short and difficult to distinguish from G2 by transcriptome, most studies combine M and G2. Therefore, we chose G0, G1, S, G2/M or G1, S, G2/M for segmentation. In **Supplementary Note 3**, we explain that different stages can be recognized by their distinctive patterns.

401 **Re. I - Figure 7** Profile of Bayes-scores and mean-scores arranged by reCAT for different clustering number k . The
 402 color bars at the bottom indicate the experimentally generated labels. The sub-order within each cluster of the color
 403 bars was arranged by the time-series after the merging process of reCAT, but before sorting out by the clustering result.
 404 The panels are mean-scores of mESC-SMARTer data for $k = 10$ (a), $k = 20$ (b) and $k = 50$ (c).

405

406 **Q15:**

407 Figure S23 only has venn plots and not ranked genes according to significance.

408 **A15:**

409 Thank you. We have added the tables.

410

411 **Q16:**

412 The authors should retain same sample names/Identifiers across main text, figures and supplementary. There are
 413 different instances of cell type, first author name, and protocol all used to represent the same sample.

414 **A16:**

415 Thank you. We have revised these issues once again.

416

417

418

419

420

421

422

423

424

425

426

427

428

429

430

431

432

-
- ¹ Kowalczyk, M.S. et al. Single-cell RNA-seq reveals changes in cell cycle and differentiation programs upon aging of hematopoietic stem cells. *Genome Res* (2015)
- ² Eisenberg, Eli et al. Human housekeeping genes, revisited. *Trends in Genetics* **29**, 569-574 (2013).
- ³ Nicholas Silver, et al. Selection of housekeeping genes for gene expression studies in human reticulocytes using real-time PCR. *BMC Molecular Biology* **7**, 1471-2199 (2006).
- ⁴ Roshan M. Kumar, et al. Deconstructing transcriptional heterogeneity in pluripotent stem cells. *Nature* **516**, 56–61 (2014).
- ⁵ Ji, Z. & Ji, H. TSCAN: Pseudo-time reconstruction and evaluation in single-cell RNA-seq analysis. *Nucleic Acids Res* **44**, e117 (2016).
- ⁶ Haghverdi, L., Buttner, M., Wolf, F.A., Buettner, F. & Theis, F.J. Diffusion pseudotime robustly reconstructs lineage branching. *Nat Methods* (2016).
- ⁷ Bendall, S.C. et al. Single-cell trajectory detection uncovers progression and regulatory coordination in human B cell development. *Cell* **157**, 714-725 (2014).
- ⁸ Sakaue-Sawano, A. et al. Visualizing spatiotemporal dynamics of multicellular cell-cycle progression. *Cell* **132**, 487-498 (2008).
- ⁹ Buettner, F. et al. Computational analysis of cell-to-cell heterogeneity in single-cell RNA-sequencing data reveals hidden subpopulations of cells. *Nature Biotechnology* **33**, 155-160 (2015).
- ¹⁰ Leng, N. et al. Oscope identifies oscillatory genes in unsynchronized single-cell RNA-seq experiments. *Nature Methods* **12**, 947-950 (2015).
- ¹¹ Kolodziejczyk, A.A. et al. Single Cell RNA-Sequencing of Pluripotent States Unlocks Modular Transcriptional Variation. *Cell Stem Cell* **17**, 471-485 (2015).

Reviewers' comments:

Reviewer #1 (Remarks to the Author):

The authors have addressed my original comments by providing a revised manuscript and a rebuttal.

A1: The authors have improved the manuscript, however, there still remain major presentational issues. In the figures, the authors should attach a label to each subplot to identify the data set used for that particular plot. The current presentation can be deceiving. For example, Figure 3a shows data from the mESC-Quartz data but Figure 3b is from the mHSC dataset. This is described in the main text but not shown in the figure or stated in the figure caption.

A3: I - Figure 1 suggests that the G1/S mean score has no discriminative power and this mainly resides in the G2/M mean score. This appears to be true throughout as the G1/S shows very little variation (similar to the housekeeping genes) and the G2/M mean score has the most variability over the cell cycle. Is cell cycle classification performance severely affected if the G1/S mean score is removed from consideration? As mentioned previously, an important part of the proposed method is its robustness, if the G1/S mean score feature is not informative then it should be removed.

A6: I - Figure 2 shows mean scores from mESC-SMARTer showing the important decrease in the G2/M toward the tail end of the cycle which is the important distinguishing factor from the other two methods. However, Bayes Scores are then shown on the mESC-Cmp data set. I think it is important that we see a similar decrease in the G2/M mean score for the mESC-Cmp data set as well to further support the authors assertion of superiority. I am concerned that a lot of this analysis still revolves around one data set (mESC-SMARTer) and that a rigorous algorithmic comparison should be made across all the labeled data sets. At the moment, it is not clear that if HMM segmentation was applied to features obtained from the TSCAN and DPT orderings would similar cell cycle classifications be given?

Additional questions:

[1] Can the authors compare their cell cycle classifications/scores from the mHSCs to those from the original study by Kowalczyk, M.S. et al. (2015)? Kowalczyk, M.S. et al. also derived cell cycle gene based scores for measuring progression.

[2] Figure S9 shows the running times of reCAT, what is the computational complexity of the entire process? How does it grow with numbers of cells? Will it process 1000s of cells which are now routinely being generated?

Overall Summary:

The paper has improved in terms of presentation and clarity but the material presented in the rebuttal should be incorporated into the main manuscript and a full set of method descriptions given. The pseudotime algorithmic comparisons should be more rigorous and the authors should make a more comprehensive, quantitative argument that their approach is robust. Although I can perfectly understand the justification, a visual check that the G2/M mean scores ordered by reCAT goes down relative to those by TSCAN and DPT on one data set is not the most convincing argument that could be made. Further work is also required to establish the robustness of the Bayes and mean scores.

Reviewer #2 (Remarks to the Author):

Review of revised manuscript from Liu et al

In the revised manuscript, Liu et al have several improvements and changes to support their computational approach 'recover cell cycle along time' (reCAT). Most of the major and minor issues have been addressed. A few points are expanded and support their claims. A few clarifications are listed here and the authors should try to address these. Overall with the revision, the manuscript and method would be useful and well utilized by the scientific community.

>>Q2: The author's benchmark reCAT on Hoechst stained mES cell-cycle dataset and further compare with human ES cell-cycle staged dataset for inconsistencies of cell-cycle marker gene expression (Figure 1). The authors should have benchmarked reCAT to more precise cell-cycle staged hES FUCCI dataset and matched with human CycleBase ranking. Subsequent reverse comparison with mESC (Buettner et al and Sasagawa et al) would better highlight the incongruence of cell cycle markers. In addition, the authors in supplementary table 1 should have rather picked top 20 ranked G2M genes and top 20 for G1- and S-phase markers from CycleBase. It is not sure if the chosen 60 genes (Supplementary table 1) are carefully selected or randomly chosen as top periodic genes (especially as the peak times seem to have poor concordance with rank).

>> A2: We did not use reCAT to process mESC-SMARTer data in Figure 1. We benchmarked reCAT on hESC data (cell-cycle staged by FUCCI) and showed the result in Re. I - Figure 4 (or Figure S10). Furthermore, FUCCI may not be able to reach enough accuracy if we want to get more subtle stage labeling⁸ and it is not as accurate as Hoechst staining for only G1, S and G2/M labeling. Genes shown in Table S1 and Table S5 are top-ranked genes, but we originally displayed them according to their peak time. In the revised manuscript, we have now rearranged the genes by 'Peakstage' and 'Rank'.

Remarks (to answers from authors): The statement by the authors is not true especially when measuring at single cell level. The combinatorial FUCCI signal intensity (Fluorescent expression marker levels) should provide extremely precise cell cycle staging of a given cell type and is definitely more accurate than Hoechst staining (DNA measurement). The errors in experimental determination of FUCCI signal intensity are likely to explain their results. Another essential point to highlight is that mRNA levels are

measured by scRNA-seq while FUCCI expression and CycleBase read out are based on protein levels (inherently more accurate).

>>Q12: The text and supplementary figures really needs to be re-ordered. The supplementary figures start from S1 then to S3, S4 further to S12 and so on making it difficult to follow the progression of work.

>>A12: We have reordered the supplementary figures. Now each supplementary figure is cited in the main text.

Remarks: The references to figures in revised manuscript still needs to be re-ordered. The first reference in text points to Fig1c and not Fig1a (Line 61)

Remarks: In Table 1, unclear what 'labeled' refers to.

Point-by-point response to the reviewers' comments

Zehua Liu, Huazhe Lou, KaikunXie, Ting Chen

We appreciate the reviewers' insightful comments, and this new revision attempts to satisfy all concerns and recommendations.

Reviewer #1

Comment 1:

The authors have improved the manuscript, however, there still remain major presentational issues. In the figures, the authors should attach a label to each subplot to identify the data set used for that particular plot. The current presentation can be deceiving. For example, **Figure 3a** shows data from the mESC-Quartz data but **Figure 3b** is from the mHSC dataset. This is described in the main text but not shown in the figure or stated in the figure caption.

Answer 1:

As suggested by the reviewer, we added a label to each subplot to identify the source dataset.

Comment 2:

Re. I - Figure 1 suggests that the G1/S mean score has no discriminative power and this mainly resides in the G2/M mean score. This appears to be true throughout as the G1/S shows very little variation (similar to the housekeeping genes) and the G2/M mean score has the most variability over the cell cycle. Is cell cycle classification performance severely affected if the G1/S mean score is removed from consideration? As mentioned previously, an important part of the proposed method is its robustness, if the G1/S mean score feature is not informative then it should be removed.

Answer 2:

Compared to the G2/M mean-score, the G1/S mean-score has two distinct features that are critical to the performance of the HMM model of reCAT. First, the G1/S mean-score has essential information to discriminate G1 from G0 stage. As shown in **Re. II - Figure 1a & 1d**, the G0 stage has much lower G1/S mean-scores than the G1 stage. Second, the peak of the G1/S mean-score generally indicates the start of the S stage, as shown in all four figures in **Re. II - Figure 1** and also in **Supplementary Note 3**. These features are taken as input to the HMM model to partition the time series into cell cycle stages. In comparison, housekeeping genes do not have such features (**Re. II - Figure 1**).

Re. II - Figure 1 Comparisons of G1/S mean-scores, G2/M mean-scores, means of log₂ expression of housekeeping genes (HK) and random selected genes (RD) using the young-MPP samples of the mHSC data (a), the E14.5 samples of the mDLM data (b), the 2i samples of the mESC-Cmp data (c) and the Mel78 samples of the hMel data (d).

Comment 3:

Re. I - Figure 2 shows mean scores from mESC-SMARTer showing the important decrease in the G2/M toward the tail end of the cycle which is the important distinguishing factor from the other two methods. However, Bayes Scores are then shown on the mESC-Cmp data set. I think it is important that we see a similar decrease in the G2/M mean score for the mESC-Cmp data set as well to further support the authors assertion of superiority. I am concerned that a lot of this analysis still revolves around one data set (mESC-SMARTer) and that a rigorous algorithmic comparison should be made across all the labeled data sets. At the moment, it is not clear that if HMM segmentation was applied to features obtained from the TSCAN and DPT orderings would similar cell cycle classifications be given?

Answer 3:

In the previous letter, we only show the Bayes-score figures of 2i samples of the mESC-Cmp data because of space limitation. In the following, we show both the Bayes-scores and mean-scores of the 2i samples in **Re. II - Figure 2a,b** (also **Figure S14a** in the manuscript), respectively. At the end of the G2/M mean-scores, we observe a very clear decrease similar to that of the G2/M Bayes-scores. Across the whole time-series, both Bayes-scores and mean-scores show a clear cyclic change, which proves the ability of reCAT to order cells accurately for the recovery of detailed gene expression variations within cell cycle stages. In comparison, we did not observe such variations in the time series produced by TSCAN, as shown in **Re. II - Figure 2c,d**.

We propose two metrics to quantitatively measure the accuracy of a time-series, leveraging the experimentally generated cell cycle stage labels, G1, S and G2/M. None of the metrics measures correctness of the order within each cell cycle stage, essentially because the cell cycle labels do not reflect such information.

We tested reCAT, TSCAN, Monocle, and DPT on the mESC-SMARTer, mESC-Quartz, and hESC datasets, and reported the results in **Re. II - Table 1**. The first metric is called correlation-score (based on Pearson’s correlation). Across all three datasets, both reCAT and TSCAN consistently outperformed Monocle and DPT, while reCAT performed slightly better than TSCAN in two out of three datasets. The second metric is called “change-index”, which measures how frequent experimentally determined single cell labels change along the time-series. Ideally, a perfect time-series would change labels twice, G1 to S and S to G2/M. Thus, we define the change-index as $1 - (s_c - 2)/(N - 3)$, where s_c is equal to the total number of label changes. A perfect time-series would have change-index value of 1, while the worst time-series where $s_c = N - 1$ would have a value of 0. The results in **Re. II - Table 1** show that reCAT outperformed the other three methods.

Although TSCAN obtains correlation-scores comparable to those of reCAT, we would like to point out that its time-series scores lower change-index values. In **Re. II - Figure 3**, we can visualize the change-index values of the hESC dataset on the color bars at the bottom of the curves. The result of TSCAN (**Re. II - Figure 3c,d**) has more frequent changes of colors compared to that of reCAT (**Re. II - Figure 3a,b**). Technically, TSCAN processes single cells by clustering and projection to capture major variations among single cells, but this approach is not sensitive enough to distinguish different cells or cell groups where subtle differences are critical to determine their order, as is evident from the rougher Bayes- and mean-score curves of the hESC data generated by TSCAN compared with those by reCAT (**Re. II - Figure 3**). Moreover, in some experimental design (i.e., the hESC dataset), cells with the same cell cycle labels were processed and sequenced in the same batch; consequently, these cells can be clustered together nicely because of batch effects, which lead to high correlation scores, but cells within each cell cycle stage may not be properly ordered.

As for the last question, the HMM segmentation developed in reCAT would not work for either TSCAN or DPT. They extract a gene set directly from input data, and this gene set varies from one dataset to another, while our HMM uses a fixed set of cell cycle genes.

Re. II - Figure 2 Comparison of the time-series by reCAT and TSCAN using the 2i samples in the mESC-Cmp data: Bayes-scores (a) and mean-scores (b) for the time-series by reCAT; Bayes-scores (c) and mean-scores (d) for the time-series by TSCAN.

Re. II - Table 1 The 'correlation-score' and 'change-index' values of the pseudo time-series by reCAT, TSCAN, Monocle and DPT, using the mESC-SMARTer, hESC and mESC-Quartz dataset.

Data sets	Methods	Correlation-score	Change-index
mESC-SMARTer	reCAT	0.80	0.70
	TSCAN	0.76	0.55
	Monocle	0.72	0.58
	DPT	0.68	0.50
hESC	reCAT	0.81	0.83
	TSCAN	0.83	0.75
	Monocle	0.52	0.73
	DPT	0.24	0.45
mESC-Quartz	reCAT	0.84	0.81
	TSCAN	0.74	0.67
	Monocle	0.63	0.48
	DPT	0.53	0.62

Re. II - Figure 3 Comparison of the time-series by reCAT and TSCAN using the labeled hESC data: Bayes-scores (a) and mean-scores (b) for the time-series by reCAT; Bayes-scores (c) and mean-scores (d) for the time-series by TSCAN.

Comment 4:

Can the authors compare their cell cycle classifications/scores from the mHSCs to those from the original study by Kowalczyk, M.S. et al. (2015)? Kowalczyk, M.S. et al. also derived cell cycle gene based scores for measuring progression.

Answer 4:

We used a method (mean-score) similar to theirs. However, we used the Cyclebase gene set, while their genes were selected from Gene Ontology and a gene list suggested by Whitfield et al (2002), followed by refining it with experimentally measured cell cycle stages. Unfortunately, we failed to find the gene list from their website.

Comment 5:

Figure S9 shows the running times of reCAT, what is the computational complexity of the entire process? How does it grow with numbers of cells? Will it process 1000s of cells which are now routinely being generated?

Answer 5:

Given n single cells and for each $k = 10, \dots, n$, reCAT computes k clusters using an EM algorithm, then finds s candidate traveling salesman cycles for these k clusters using the arbitrary insertion algorithm and choose the shortest route among the s cycles. This is because the arbitrary insertion algorithm is a randomized algorithm, and it returns a different solution each time. Then, all these chosen traveling salesman cycles corresponding to $k = 10, \dots, n$ were merged into a consensus cycle of n single cells. The time complexity for computing k clusters is $O(nkt)$, where t is the number of iterations in EM, and the time complexity for ordering these k clusters into traveling salesman cycles is $O(sk^2)$. Thus the overall time complexity is $O(n^3(t + s))$.

The current version of reCAT was implemented in R and it takes about 20 minutes to process 300 cells. For 1000 cells, it will take about 10 hours. We plan to re-implementation reCAT using C++, thus enabling it to process thousands of cells efficiently.

Comment 6: (Summary)

The paper has improved in terms of presentation and clarity but the material presented in the rebuttal should be incorporated into the main manuscript and a full set of method descriptions given. The pseudo time algorithmic comparisons should be more rigorous and the authors should make a more comprehensive, quantitative argument that their approach is robust. Although I can perfectly understand the justification, a visual check that the G2/M mean scores ordered by reCAT goes down relative to those by TSCAN and DPT on one data set is not the most convincing argument that could be made. Further work is also required to establish the robustness of the Bayes and mean scores.

Answer 6:

We have made incorporations in the revised manuscript. In **Q3**, we gave detailed and rigorous comparisons of state-of-the-art pseudo time algorithms, even though lacking ground-truth datasets to measure the accuracy of single cell order within cell cycle stages. The robustness of our TSP algorithm was tested in the following way. Given n labeled single cells, we compute K clusters, run our TSP algorithm 200 times to produce 200 solutions for each K , compute their correlation scores, and plot score distributions. The plots of the labeled hESC, mESC-Quartz and mESC-SMARTer datasets are shown in

Re. II - Figure 4a (hESC),

Re. II - Figure 4b (mESC-Quartz), and **Figure 2c** (mESC-SMARTer), respectively. For all these plots, the correlation-scores are stable and have low variations, showing robustness of the solutions given by reCAT.

Both the Bayes-score and mean-score methods have valid theoretical bases. In fact, we adopted the mean-score approach from Kowalczyk, M.S. et al. (2015). This research group also applied this computational method to multiple scRNA-seq datasets from six cell lines in three subsequent scRNA-seq studies^{1,2,3}, in which

the method exhibited strong cell cycle signals, leading to novel biological discoveries. Hence the robustness of the mean-score method has been proved in these applications.

The Bayes-score method adopted a commonly used feature selection method for cell classification^{4,5,6}. We demonstrated that the Naïve Bayes model was more robust to noise than other computational methods such as the Logistic regression (**Supplementary Note 2**). We trained and tested the Bayes-score method on the labeled mESC-SMARTer dataset. Through ten-fold cross-validation, we found that 94.3% of the G1 cells had higher G1 Bayes-scores than the G2/M Bayes-scores, and that 92.9% of the G2/M cells had higher G2/M Bayes-scores than G1 Bayes-scores. The results proved that the Bayes-scores were very accurate in discriminating between G1 and G2/M cells. We then applied the trained Bayes-score method to discriminate G1 and G2/M cells on the labeled hESC and mESC-Quartz datasets. The results of the hESC dataset showed that the error rates were 18.4% for the G1 cells and 36.8% for the G2/M cells. The results of the mESC-Quartz dataset showed that all the G1 and G2/M cells were correctly separated. We did not test the model on the S stage because the S stage is a transitive stage, which can easily be confused with the G1 and G2/M stages. However, the HMM in reCAT can combine all Bayes-scores and mean-scores together to segment the pseudo time-series into different cell cycle stages.

Re. II - Figure 4 Robustness of the TSP algorithm in reCAT using labeled datasets, hESC (a) and mESC-Quartz (b), in which the horizontal axes represent cluster number 'K', and a half width of the error bars stands for a standard deviation of the correlation-scores.

Reviewer #2

Comment 1:

The statement by the authors is not true especially when measuring at single cell level. The combinatorial FUCCI signal intensity (Fluorescent expression marker levels) should provide extremely precise cell cycle staging of a given cell type and is definitely more accurate than Hoechst staining (DNA measurement). The errors in experimental determination of FUCCI signal intensity are likely to explain their results. Another essential point to highlight is that mRNA levels are measured by scRNA-seq while FUCCI expression and CycleBase read out are based on protein levels (inherently more accurate).

Answer 1:

We revised our sentences in the following. Hoechst staining is a fundamental approach for determining cell cycle stage, and FUCCI could potentially produce cell cycle labels with a higher resolution. We have revised related presentation in the manuscript. Thank you!

Comment 2:

The references to figures in revised manuscript still needs to be re-ordered. The first reference in text points to Fig1c and not Fig1a (Line 61). In Table 1, unclear what 'labeled' refers to.

Answer 2:

We have revised the problems. Thank you!

¹ Macosko, E. Z. et al. Highly parallel genome-wide expression profiling of individual cells using nanoliter droplets. *Cell* **161**, 1202–1214 (2015).

² Tirosh, I. et al. Dissecting the multicellular ecosystem of metastatic melanoma by single-cell RNA-seq. *Science* **352**, 189–196 (2016).

³ Tirosh, I. et al. Single-cell RNA-seq supports a developmental hierarchy in human oligodendrogloma. *Nature* **539**, 309–313 (2016)

⁴ Geman, D. et al. Classifying gene expression profiles from pairwise mRNA comparisons. *Stat. Appl. Genet. Mol. Biol.* **3**. Article 19. (2004)

⁵ Aik Choon Tan, A. C. et al. Simple decision rules for classifying human cancers from gene expression profiles. *Bioinformatics* **21** (20), 3896–3904. (2005)

⁶ Afsari, B. et al. Switchbox: an r package for k-top scoring pairs classifier development. *Bioinformatics* **31** (2), 273–274. (2015)

Reviewers' comments:

Reviewer #1 (Remarks to the Author):

The authors have addressed the questions I raised in the previous review. However, a few key issues still require clarification:

"First, the G1/S mean-score has essential information to discriminate G1 from G0 stage." -- this response is problematic because in the previous rebuttal, the authors produced a figure that did not contain any G0 cells and now in this rebuttal the argument changes to include G0 cells. Nonetheless, the authors response would imply that my comment is correct, the G1/S score does not help much in indiscriminating between G1/S and G2/M cells. The G2/M score is the most important feature. Whilst I agree the G1/S score might help to identify G0 is this not simply due to the fact that in a quiescent state, the cells generally tend to have low expression?

"Second, the peak of the G1/S mean-score generally indicates the start of the S stage, as shown in all four figures in Re. II - Figure 1 and also in Supplementary Note 3." -- Please mark the positions on the plots where the authors believe this score peaks in the figures. It is not obvious at all that it does "peak" and I suspect most readers would struggle to visualise this too.

"Although TSCAN obtains correlation-scores comparable to those of reCAT, we would like to point out that its time-series scores lower change-index values." -- the performance difference is not obviously that much different especially when you consider that TSCAN is itself not tuned for cyclical processes. reCAT would suffer if the quantitative scoring index reflected the fact that some G2 cells were placed amongst the G1s, etc.

"Unfortunately, we failed to find the gene list from their website." -- email the authors? Furthermore my specific question was how well the cell cycle stage classifications match up between reCAT and the original study. This can be done whether it is on the same gene set or not. It is not a true performance measure but more a sense of whether reCAT produces consistent classifications or if it derives something completely different.

Overall, the driver for my questions above stems from the fact that this method (if published) has the potential to be very well-used in the single cell community. Therefore establishing the robustness of the method is critical.

Reviewer #2 (Remarks to the Author):

Review of revised manuscript from Liu et al

In the second revision of the manuscript, Liu et al have made improvements and necessary changes to support the claims for their computational approach 'recover cell cycle along time' (reCAT). The major and minor issues have been addressed including presentation issues and clarifying sections within text.

Overall with the revision, the manuscript and method would be useful to the scientific community.

Point-by-point response to the reviewers' comments

Zehua Liu, Huazhe Lou, KaikunXie, Ting Chen

We appreciate the reviewers' insightful comments, and this new revision attempts to satisfy all concerns and recommendations.

Reviewer #1

Comment 1:

"First, the G1/S mean-score has essential information to discriminate G1 from G0 stage." -- this response is problematic because in the previous rebuttal, the authors produced a figure that did not contain any G0 cells and now in this rebuttal the argument changes to include G0 cells. Nonetheless, the authors response would imply that my comment is correct, the G1/S score does not help much in indiscriminating between G1/S and G2/M cells. The G2/M score is the most important feature. Whilst I agree the G1/S score might help to identify G0 is this not simply due to the fact that in a quiescent state, the cells generally tend to have low expression?

Answer 1:

We generally agree with the reviewer. In **Reply Letter I**, the dataset (mESC-SMARTer) did not contain G0 cells. However, **Reply Letter II** showed figures of other datasets with G0 (young-MPP in mHSC and Mel78 in hMel) to offer a more comprehensive explanation. For the last question, it is true that cells generally have low expression in G0 stage, and this is why we normalized expression levels in the preprocessing step.

Comment 2:

"Second, the peak of the G1/S mean-score generally indicates the start of the S stage, as shown in all four figures in Re. II - Figure 1 and also in Supplementary Note 3." -- Please mark the positions on the plots where the authors believe this score peaks in the figures. It is not obvious at all that it does "peak" and I suspect most readers would struggle to visualize this too.

Answer 2:

We have revised the figures in the following.

Re. III - Figure 1 Comparisons of G1/S mean-scores, G2/M mean-scores, means of log₂ expression of housekeeping genes (HK) using the young-MPP samples of the mHSC data (a), the E14.5 samples of the mDLM data (b), the 2i samples of the mESC-Cmp data (c) and the Mel78 samples of the hMel data (d). The start of the S stage is marked by dashed.

Comment 3:

"Although TSCAN obtains correlation-scores comparable to those of reCAT, we would like to point out that its time-series scores lower change-index values." -- the performance difference is not obviously that much different especially when you consider that TSCAN is itself not tuned for cyclical processes. reCAT would suffer if the quantitative scoring index reflected the fact that some G2 cells were placed amongst the G1s, etc.

Answer 3:

Cyclical consideration is one of the features that distinguishes reCAT from other methods, and we agree that incorrectly labeled cells would reduce the effectiveness of quantitative scores.

Comment 4:

"Unfortunately, we failed to find the gene list from their website." -- email the authors? Furthermore, my specific question was how well the cell cycle stage classifications matchup between reCAT and the original study. This can be done whether it is on the same gene set or not. It is not a true performance measure but more a sense of whether reCAT produces consistent classifications or if it derives something completely different.

Answer 4:

We could not get the original gene list from the authors but they sent us another list, which partially overlaps with our list, as shown in **Re. III - Figure 2a**. We calculated the mean-scores of the two lists using all the cells in the mHSC dataset (about 1150). Then we performed a linear regression of G1/S-scores and G2/M-scores between the two groups, and obtained r^2 values of 0.904 and 0.853, respectively. The high correlations indicate that both gene lists work well for reCAT.

Next, we examined whether reCAT produces consistent classifications using these two gene lists. We chose the multi-potent progenitor samples from young mice (young-MPP) in the mHSC dataset, and ran reCAT to

order and segment the samples into G0, G1, S, and G2/M stages, as shown in **Re. III - Figure 2b, c**. The results are highly consistent: only one cell, which was the last G1 cell before S using the Cyclebase gene list, was classified as S using the provided gene list. The segmentation proportions are also consistent with that of the original study¹.

Re. III - Figure 2 (a) Overlap between the reCAT used gene list (Cyclebase) and the provided gene list. (b) The segmentation result of the young-MPP data generated by reCAT using the mean-scores of the Cyclebase genes. (c) The segmentation result of the young-MPP data generated by reCAT using the mean-scores of their provided genes. We marked the peak area of the regressed G1/S curves, which indicate the start area of S stage.

Reviewer #2

Comment 1:

In the second revision of the manuscript, Liu et al have made improvements and necessary changes to support the claims for their computational approach 'recover cell cycle along time' (reCAT). The major and minor issues have been addressed including presentation issues and clarifying sections within text. Overall with the revision, the manuscript and method would be useful to the scientific community.

Answer 1:

Thank you!

¹ Kowalczyk, M.S. et al. Single-cell RNA-seq reveals changes in cell cycle and differentiation programs upon aging of hematopoietic stem cells. *Genome Res* (2015).

REVIEWERS' COMMENTS:

Reviewer #1 (Remarks to the Author):

The authors have adequately addressed my previous questions. It would be useful to include some of the responses into the text (such as Answer 3) to address the limitations of the metrics.

Point-by-point response to the comments

Zehua Liu, Ting Chen

Reviewer #1

Comment 1:

The authors have adequately addressed my previous questions. It would be useful to include some of the responses into the text (such as Answer 3) to address the limitations of the metrics.

Answer 1:

Thank you! We have added the mentioned content into the manuscript.